# A Comprehensive Review on Adsorption, Photocatalytic and Chemical Degradation of Dyes and Nitro-Compounds over Different Kinds of Porous and Composite Materials

**DOI:** 10.3390/molecules28031081

**Published:** 2023-01-21

**Authors:** Abdul Haleem, Anum Shafiq, Sheng-Qi Chen, Mudasir Nazar

**Affiliations:** 1School of Chemistry and Chemical Engineering, Jiangsu University, Zhenjiang 212013, China; 2School of Mathematics and Statistics, Nanjing University of Information Science and Technology, Nanjing 210044, China; 3Jiangsu International Joint Laboratory on System Modeling and Data Analysis, Nanjing University of Information Science and Technology, Nanjing 210044, China; 4Anhui Province Key Laboratory of Pharmaceutical Preparation Technology and Application, Key Laboratory of Xin’an Medicine, The Ministry of Education, Anhui University of Chinese Medicine, Hefei 230038, China; 5Biofuels Institute, School of the Environment and Safety Engineering, Jiangsu University, Zhenjiang 212013, China

**Keywords:** organic pollutants, polymeric materials, porosity, carbon materials, composite, metal–organic frameworks, layered double hydroxides, biosorbents

## Abstract

Dye and nitro-compound pollution has become a significant issue worldwide. The adsorption and degradation of dyes and nitro-compounds have recently become important areas of study. Different methods, such as precipitation, flocculation, ultra-filtration, ion exchange, coagulation, and electro-catalytic degradation have been adopted for the adsorption and degradation of these organic pollutants. Apart from these methods, adsorption, photocatalytic degradation, and chemical degradation are considered the most economical and efficient to control water pollution from dyes and nitro-compounds. In this review, different kinds of dyes and nitro-compounds, and their adverse effects on aquatic organisms and human beings, were summarized in depth. This review article covers the comprehensive analysis of the adsorption of dyes over different materials (porous polymer, carbon-based materials, clay-based materials, layer double hydroxides, metal-organic frameworks, and biosorbents). The mechanism and kinetics of dye adsorption were the central parts of this study. The structures of all the materials mentioned above were discussed, along with their main functional groups responsible for dye adsorption. Removal and degradation methods, such as adsorption, photocatalytic degradation, and chemical degradation of dyes and nitro-compounds were also the main aim of this review article, as well as the materials used for such degradation. The mechanisms of photocatalytic and chemical degradation were also explained comprehensively. Different factors responsible for adsorption, photocatalytic degradation, and chemical degradation were also highlighted. Advantages and disadvantages, as well as economic cost, were also discussed briefly. This review will be beneficial for the reader as it covers all aspects of dye adsorption and the degradation of dyes and nitro-compounds. Future aspects and shortcomings were also part of this review article. There are several review articles on all these topics, but such a comprehensive study has not been performed so far in the literature.

## 1. Introduction

On this planet, water is vital for the existence of different biota. Maintaining the present water quality is a global demand that increases daily. Unfortunately, disasters, conflicts, and wars over inadequate water resources are commonplace. Along with this problem, different kinds of pollutants in water are also the main reason for the declination of water specifications [1]. Among all these pollutants, dyes and nitro-compounds are the main pollutants that highly affect water sources, worsening the quality of life of aquatic organisms. The rapid industrialization of various dyes and nitro-compounds has affected the global economy. Different industries, such as the fiber, pharmaceutical, and cloth industries, are the main reasons for different kinds of pollution, such as water, soil, and air. Among all the organic pollutants, dyes and nitro-compounds are the critical parts of this review article.

### 1.1. Dyes

Dyes are the main responsible pollutant among all others, extensively implemented in the food, paper, textile, and pharmaceutical industries with a yearly production surpassing 7 × 105 tons [2,3]. Dye production in the United States, western Europe, and Japan has considerably diminished, though its production in India, China, and a few South Asian countries have enhanced over the past 25–30 years [4]. China is the most significant textile producer, representing almost 55% of global textile consumption. Furthermore, China and India are large exporters of various dyes and dye-intermediate chemicals [4]. The lower solubility and superfluous stability of dyes make them persistent and threaten aquatic organisms’ life [5]. The textile, chemical, and pharmaceutical industries mainly produce hazardous and unwanted chemicals that directly influence animals and plants [6]. Textile industries are a leading source of pollution for water reservoirs because they use a massive amount of water and more than 8000 chemicals. Literature studies show that for every 8000 kg of cloth produced, almost 1.6 million liters of water are used daily. In textile industries, the large amounts of water used during the wetting process produce a massive amount of soluble dyestuff in wastewater; an alarming sign for aquatic organisms [7]. The World Bank Report highlights that textile industries are highly responsible for water pollution (17% to 20% of total water pollution) [8].

Similarly, food industries also use a large number of different dyes for the improvement of texture and attractive qualities. High solubility and cheap rates make synthetic dyes an attractive candidate in food industries. In the industrial zone, more than 10,000 dyes are used for numerous purposes. The annual production of all these synthetic dyes is estimated to be about 30,000 tons [9,10,11]. All these dyes have a complex chemical construction which makes them reasonably unassailable to degradation, and hence remain in the environment for a longer time, making it harder for light to enter into water bodies, and, therefore, affecting the normal process of photosynthesis of aquatic flora.

In the same way, dyes also highly affect the life of animals. The dyes are also responsible for changing conditions, such as biological oxygen demand (BOD), chemical oxygen demand (COD), dissolved oxygen concentration, and pH. Among all the synthetic dyes, some possess carcinogenic properties [12,13]. Dyes discharge into water bodies, reducing light penetration and affecting the performance of algae and aquatic plants.

Furthermore, the dyes ingested by the fishes and other aquatic organisms can produce toxic intermediates after metabolism and negatively impact the health of fishes and their predators [1]. Along with aquatic organisms, humans and other mammals can be exposed to these dyes through oral ingestion or skin contact. Intestinal microflora in the human gut convert these dye molecules into toxic amino acids, which harm the various tissues in the human body [14,15]. Bacteria cultured from human skin were also able to degrade dye molecules and produce carcinogenic amines [15,16]. The primary diseases caused by dyes in fish are gill diseases (disrupting typical architecture, secondary lamellae fusion, telangiectasia, epithelial uplifting, micronuclei, nuclear buds, fragmented apoptosis, binucleated cells, aneurism, and hyperplasia, etc.), liver diseases (disrupting standard architecture, infiltration of lymphocytes, sinusoidal dilations, necrosis, and vacuolation, etc.), kidney diseases (renal tubule degeneration, reduced kidney lumen, etc.), gut diseases (histopathological alteration and dysbiosis of gut microbiota), and cell diseases (disturb several enzymatic activities, accelerate lipid peroxidation, enhancing DNA damage, change in red blood cell (RBC) shape and size, etc.) [16,17]. Along with fish diseases, dyes also affect the human-like central nervous system (inhibiting intracellular enzymes of the central nervous system and damage of the central nervous system), liver (hepatocarcinoma, increased serum alkaline phosphate and gamma–glutamyl transferase levels, and liver damage), kidney (reticular cell sarcoma, kidney damage, and bladder cancer), skin (dermatitis, allergic conjunctivitis, rhinitis, occupational asthma, and other allergic reactions), enzymatic system (inactivation of enzymatic activities, carcinogenic aromatic amines formation, block enzymes including glutathione reductase and disturb cellular redox equilibrium), human chromosomes (strong genotoxic effect, intercalating with the helical structure of DNA and duplex RNA, mutagenic potentiality and carcinogenic agents) and reproductive system (cytotoxic effect on spermatozoa cells, testes weight reduction, decline in ovarian protein and glucose) [17].

There are three main categories of dyes: (a) anionic dyes, mainly with an SO^−3^ group present in their primary structure; (b) cationic dyes, mainly due to the presence of a protonated amide group; and (c) nonionic dyes, according to their dissociation behavior in aqueous solutions [18]. Azo-based dyes (anionic or cationic) have one extra azoic bond, which gives more stability to the respective dyes against heat, light, and aerobic digestion, and may be a severe cause of vomiting, allergic problems, cyanosis, and genetic mutation. The dye industries give characteristic examples of the potential influence of unrestrained management of polluted streams [19]. The spreading of dyes in water bodies highly affects the solubility of gas, which in turn affects the gills of water-based organisms and their breeding places and shelters. The dispersion of dyes in water bodies also hinders light penetration, which highly affects photosynthesis. Along with these secondary effects, dyes have specific noxiousness due to their toxic and mutagenic effects [20,21]. With the above-mentioned adverse effect of dyes on the environment in mind, it is vital to remove or degrade them from water bodies to keep aquatic organisms, animals, and plants safe and healthy. Table 1 displays different types of dyes and their description and applicability.

### 1.2. Nitro-Compounds

The other main group of pollutants is nitro-compounds, a valuable intermediate for various industrial uses. Nitro-compounds with an aromatic base are mainly starting materials for commercial uses, but their aliphatic types display a more diverse chemical behavior under reducing circumstances [22]. Nitro-compounds are highly toxic and the primary source of water pollution. Among the nitro-compounds, phenol-based organic compounds (2-nitrophenol and 4-nitrophenol), nitrobenzene, and nitro-aniline, referred to as “priority organic pollutants” due to their excellent solubility and structural stability in water, have gained significant attention from environmental researchers in the 21st century [23]. Each nitro-compound has at least one nitro group in its structure. These compounds have a unique affinity for electrons and reduction potential; such compounds act like electron donor material. The above properties stabilize the nitro-compounds against oxidative degradation [24]. Nitrophenol (NP) is the essential raw material for fabricating textiles, dyes, paper, pharmaceuticals, pesticides, etc. At the same time, dinitrophenol (DNP) and trinitrophenol (TNP) are extensively used as explosive materials in industries and are usually identified as explosophore. Therefore, in the industries mentioned above, different forms of disposal upsurge the growth of the noxious contaminants, highly affecting the environment [25]. In the environmental and several other scientific fields, nitro-group-containing organic compounds play a vital role due to their double-edged nature, seeing continuous usage in the field of therapeutics, though still considered as a toxicophore in nature. They are mainly responsible for causing carcinogenicity, mutagenicity, genotoxicity, and hepatotoxicity.

A short-time exposure to NP in humans can cause irritation of the eyes, mucous membranes and respiratory tract, vomiting, headache, nausea, and fatigue. Long-term NP exposure can highly affect the peripheral nervous systems, central nervous systems, liver, and kidneys [26]. In the same way, nitrobenzene (NB) is highly responsible for causing conditions such as skin irritation, anemia, and cancer. In humans, NB mainly causes noticeable methemoglobin formation, neurotoxic effects, cyanosis, gastric irritation, unconsciousness, vomiting, nausea, cyanosis, seizures, drowsiness, coma, and, finally, respiratory failure ending in death [27]. Due to the higher toxicity of nitro-compounds, the USA Environmental Protection Agency (EPA) registered them as the highest-priority toxins. Consequently, their removal from the environment is vital. New advancements in science have made it possible to overcome the toxic effects of these nitro-compounds while favoring the valuable ones [28].

Therefore, it is also vital to handle nitro-compounds carefully before disposing of them in water bodies as they can affect the health of both aquatic organisms and humans Three main strategies are used (adsorption, photocatalytic degradation in the presence of sunlight, and chemical degradation using a reducing agent) to remove and degrade the above-mentioned organic pollutants before they enter water bodies. The three strategies are discussed in detail, as shown in Figure 1.

## 2. Dye Adsorption and Kinetics

Adsorption is the most clean and friendly physical method for removing dyes from water bodies to save the lives of aquatic organisms and make the environment clean for human beings. In this method, dyes are absorbed on the surface of fabricated materials, and one can monitor this process via a UV-visible spectrometer at particular time intervals. Initially, the dye concentration can be checked via a spectrophotometer. After a particular time, the absorption peak will decrease to the minimum, which is the indication of dye adsorption, and the color will also be near-colorless depending on the nature of the dye, because each dye has a specific color due to a specific chromophore. It can also be confirmed by Fourier-transform infrared spectroscopy (FTIR) and X-ray photon spectroscopy (XPS) techniques. During the adsorption of dye molecules, a noticeable change will be observed in both FTIR and XPS peaks, which will be the primary indication of dye adsorption over a specific sorbent. The adsorption capacity of dye and the percentage removal can be calculated using the following Equations (1) and (2), respectively:(1)$$q_{t}=\frac{\left(c_{0}-c_{a}\right)\times V\times M}{m_{0}}$$
(2)$$Removal\;\left(\%\right)=\frac{\left(c_{0}-c_{e}\right)\;}{c_{0}}$$

In the above two equations, *c*_0_ and *c*_e_ stand for the initial and equilibrium concentration of the adsorbed dye, respectively. *c*_a_ stands for the dye concentration at a certain point. *M* (g/mol) stands for the dye’s relative molecular weight. *V* (L) stands for the volume of the dye solution. *m*_0_ (g) stands for the dry weight of the material.

Three adsorption kinetics have been studied so far for the adsorption of dyes over different kinds of materials: pseudo-first-order, pseudo-second-order, and intra-particle diffusion equations. Among these equations, the pseudo-first-order equation (Equation (3)) is mainly used to examine the dynamic adsorption performance of dye. This equation examines the liquid–solid system adsorption behavior to reflect the adsorption rate [29]:(3)$$\ln\left(q_{e}-q_{t}\right)=lnq_{e}-K_{1}t$$

In Equation (3), *q_t_* (mg/g) and *q*_e_ (mg/g) stand for the amount of dye adsorption at *t* time and the adsorption amount at equilibrium, respectively. *K*_1_ stands for the rate constant of pseudo-first-order adsorption [30].

The pseudo-second-order model (Equation (4)) was also established, which depends on the hypothesis that the adsorption capability is mainly proportional to the number of active sites occupied on the adsorbent (material) [31]:(4)$$\frac{t}{q_{t}}=\frac{1}{K_{2}q_{e}^{2}}+\frac{t}{q_{e}}$$

In Equation (4), *K*_2_ stands for the rate constant of pseudo-second-order adsorption.

The intra-particle diffusion model (Equation (5)) can also be cast off to illustrate the control steps in the adsorption process by identifying the adsorption mechanism’s diffusion mechanism:(5)$$q_{t}=K_{p}t^{1/2}+C$$

In Equation (5), *K*_*p*_ stands for the intra-particle diffusion rate constant, and *C* (mg/g) stands for the constant.

Furthermore, dye adsorption has always been studied by the two essential adsorption isotherms; the Langmuir equilibrium isotherm and Freundlich equilibrium isotherm, as mentioned in Equation (6) and Equation (7), respectively [32].
(6)$$\frac{C_{e}}{Q_{e}}=\frac{C_{e}}{Q_{m}}+\frac{1}{K_{L\;}Q_{m}}$$

In Equation (6), *C*_*e*_ stands for equilibrium concentration (mg/L), *Q*_*e*_ stands for equilibrium capacity of adsorption (mg/g), *Q*_*m*_ stands for the maximum capacity of adsorption (mg/g), and *K*_*L*_ stands for Langmuir constant (L/mg).
(7)$$R_{L}=\frac{1}{1+K_{L}C_{0}}$$

In Equation (7), *C*_0_ stands for initial dye concentration (mg/L), and *K*_*L*_ stands for Langmuir constant (L/mg).

The Langmuir adsorption isotherm follows the monolayer formation of adsorbent over adsorbate, supposing that adsorption takes place on the homogeneous solid surface of identical sites with equivalent energy. On the other hand, the Freundlich adsorption isotherm considers that multilayer adsorption occurs with heterogeneous energy distribution of active sites, accompanied by the interaction between different adsorbed molecules, as previously explained in many books and review papers. Here, we will mainly focus on the materials used for the adsorption studies and the adsorption mechanism.

### 2.1. Materials Used for Dye Adsorption

This review article mainly focused on six different materials: polymers, carbon-based materials, clay-based materials, metal–organic framework (MOF) materials, layered double hydroxide (LDH) materials, and biosorbents for the adsorption of dyes. We elaborated on each in detail, as shown in Figure 2. In adsorption, a pristine material is needed in which different functional groups are available for the adsorption of dyes. Other factors, such as porosity, temperature, pH, etc., are also responsible for the efficient adsorption of dyes.

### 2.2. Pure Polymeric Materials for Dye Adsorption

Polymer is a material with a high affinity for dye removal due to its three-dimensional (3D) structure and multiple functional groups in its network. Diverse categories of polymers (microgels, hydrogels, magnetic hyperbranched polymers, conducting polymers, and cryogels) have been used so far for the adsorption of different water-soluble organic dyes. Here, we will shortly explain which polymers are more suitable for dye adsorption and why [33]. The adsorption of dyes takes place in all kinds of polymeric materials through electrostatic interaction, hydrogen bonding, and Van der Waals forces. It is well-known that polymeric materials have different functional groups in their 3D networks (-COOH, -NH_2_, -SO_3_H, -OH, -C=O, etc.), which mainly interact with different functional groups of complex dye structures, as shown in Figure 3 [2]. In Figure 3, different kinds of interaction, such as chemisorption (electrostatic interaction and hydrogen bonding) and physical adsorption of dyes occur, depending on the nature of the dyes. Along with the importance of functional groups in polymeric materials, porosity is the main parameter that decides the fast and efficient adsorption of dyes from water bodies [2]. The porosity of the polymeric materials is directly proportional to the synthetic route implemented for the polymer synthesis. For the synthesis of most polymeric materials, a conventional free radical polymerization route is implemented (65–70 °C or room temperature). Conventional polymeric materials (microgels, hydrogels, hyper-branched polymers, and conducting polymers) have been used for a few decades for the adsorption of organic water-soluble dyes and other pollutants. Still, their closed cavities make them an unideal candidate for adsorption studies [2,34,35]. Their adsorption efficiency is not bad and quite satisfactory, but faster adsorption rate is required in this computerized world.On the other hand, cryogels are the most important candidate for removing dyes and will be useful in wastewater treatment over conventional polymeric hydrogels. Cryogels have macroporous structures due to their synthetic route (cryogenic conditions), making these materials more suitable for different applications already studied by our group and other researchers [36,37,38,39,40,41,42,43,44,45,46], as if the materials have a better adsorption capacity but a closed cavity, they will not be considered a suitable candidate for adsorption studies due to their long duration of adsorption. Cryogels have a pore size in the 50–200 um range, facilitating dyes’ fast diffusion and adsorption. Polymeric materials have different functional groups, such as amide, carboxylic, sulfonic, hydroxyl, etc., which are mainly responsible for the adsorption of various dyes [47]. The adsorption of dye is also highly affected by temperature and pH. Cationic dyes will rapidly and efficiently adsorb anionic dyes and vice versa. Polyacrylamide cryogel was used for the adsorption of orange G (OG) dyes, and adsorption equilibrium was obtained in 30 min. Such fast adsorption is due to the large pores of cryogels fabricated at cryogenic conditions. Cryogels are superior to conventional hydrogels in their structural flexibility, mechanical stability, and fast reaction capability. Conventional hydrogels were also used for dye adsorption, but due to their closed cavity and dense structure, they were not the best choice for dye adsorption. Temperature and pH are highly responsible parameters for the adsorption of dyes over different polymeric materials. Yin et al. reported sodium alginate/poly(sodium p-styrene sulfonate) (AlgMA/PNaSS) cryogels for the selective adsorption of rhodamine B (RhB), new coccine (NC), methyl orange (MO), crystal violet (CV) and methylene blue (MB) dyes [48]. The adsorption of dyes depends on various parameters already explained by these researchers, but the dye size is also responsible for dye adsorption over cryogels. MB, RhB, and CV dyes had a high adsorption capacity on cryogels due to the presence of positive charges in their network, and electrostatic interaction will occur between them. The adsorption capabilities of AlgMA/PNaSS cryogels for crystal violet and rhodamine B were lower than that of methylene blue. It can be clarified that rhodamine B and crystal violet have bigger molecular sizes than methylene blue. The adsorption behavior happened on the surface or in the large pores of the cryogels, but it was hard for them enter into the small pores of the cryogels [49]. Cryogels have been extensively studied by many researchers for their adsorption of different cationic and anionic dyes [47,50,51,52,53]. From the extensive literature study, it is obvious that cryogels are very useful candidates for the adsorption and removal of various organic dyes and pollutants to ensure environmental stability, due to the macro-porous structure of cryogels and the cryogenic conditions which are mainly responsible for this macro-porosity.

### 2.3. Carbon-Based Composite Materials and Biochar for Dye Adsorption

Carbon-based materials are valuable candidates for the adsorption of dyes, metal ions and various antibiotics due to their large surface area, thermal and mechanical stability, and multiple functionalities in their network. In the previous decade, the adaptation of new carbon materials, such as diamonds, carbon nanotubes (CN), carbon dots, and graphene oxide (GO) have been shown to be valuable materials for adsorption studies [54]. GO is a 2D material with multiple oxygen moieties in its network that is mainly used for the selective adsorption of cationic dyes. GO can adsorb methylene blue (MB) quickly from a mixture of methylene blue (MB) and methyl orange (MO). The selectivity was explained by an ab initio method, where it was found that MB adsorption energy on GO was higher in comparison with MO (−2.25 vs. −1.45 eV), which confirmed much stronger adsorption of MB on a GO sheet than MO [55]. For more adsorption of dyes, different kinds of doped composites were designed by the researchers, showing better adsorption efficiency than pure GO-based material. Acrylic acid doped on GO can adsorb 100% MB at the expense of 10% MO [56].

Similarly, the researchers also fabricated boron-doped GO and 3D GO-based aerogels with better adsorption capacity for different dyes [57]. Different carbon nanotubes and their sponges were fabricated along with GO to remove extra water-soluble organic dyes [54]. The adsorption of dyes depends on the nature of the functional groups present in the materials and on the nature of dyes (anionic or cationic). If the material contains more negative charges, it will have adsorbed cationic dyes, and vice versa. Different kinds of interaction (hydrogen bonding, van der Waals interactions, and London forces) are responsible for dyes’ adsorption, as shown in Figure 4 [34]. As shown in Figure 4, the dye molecules are rich in sulfonic groups and aromatic rings, which offers strong electrostatic interaction as well as π-π stacking between molecules of dyes and the fabricated spongy materials. This interaction is further boosted by hydrogen bonding. The combination of all these interactions makes the material superior for the uptake of dyes [58]. Carbon nanotubes (CNTs) are promising adsorbing materials for dye adsorption. Researchers developed various CNTs based on activated carbon for the removal and adsorption of dyes and other heavy metals from water bodies due to their large surface area, stability, easy modification, and high strength [59,60]. Pure CNTs have less adsorption capacity for methylene blue than GO; CNTs have a larger surface area than GO, as already reported by Li et al. [61]. The main reason for the low adsorption of CNTs is their cylindrical structure, which mainly hides the smooth adsorption of dye molecules through electrostatic interaction compared to GO sheets, which have multiple oxygen moieties for interaction with dye molecules [61]. The adsorption of dyes can be improved by the hybrid structure of CNTs and GO, as previously reported [62,63,64]. Carbon-based materials are a promising candidate for the adsorption of dyes from water bodies to make the environment clean and suitable for living organisms.

Biochar is another promising material for the adsorption of organic pollutants and heavy metals due to its porous nature, large surface area, and multiple functional groups [65,66,67]. Biochar is a solid carbonaceous product of biomass carbonization in a vacuum, or the presence of significantly less oxygen, which has received much attention due to its carbon sequestration, waste recycling, and energy production in recent years [68,69]. For biochar production, primarily biomass feedstock sources such as forestry waste, agricultural waste, and animal manure are used [70]. Amongst all the waste biomass feedstock, rice straw for biochar production is an ideal source because rice straw accounts for about 731 million tons globally (90% in Asia alone) [71]. Numerous researchers have worked on different sources for biochar production, such as rice straw, Korean cabbage, wood chip, pine wood biochar, pig manure biochar, paper-derived biochar, peels of oranges, peels of bananas, coir pith, date pith, peanut shells, almond hulls, Citrullus lanatus rind, ZnO/cotton stalk biochar, cow dung biochar, litchi peel biochar, sludge biochar, low-rank coal (leonardite), coffee waste-activated biochar, oil palm waste biochar, waste palm shell biochar, wet-torrefied microalgal biochar, and green biochar/iron oxide, for the adsorption of various dyes (congo red, crystal violet, methylene blue, aid violet, basic blue 9, reactive dye, acid yellow 36, rhodamine B, malachite green, etc.) [72,73,74,75,76,77,78,79,80,81]. From the thermodynamic parameters as reported by Zhang et al., physical sorption is the primary adsorption mechanism through which methylene blue adsorbs biochar/Fe_x_O_y_ [80]. This was further verified by Zhang et al. through an XPS survey that electrostatic interactions also occur during the adsorption of dye molecules over the biochar. Overall, all of these carbon-based materials are promising adsorbing candidates for removing dyes from water bodies to make them environmentally friendly.

### 2.4. Clay-Based Composite Materials for Dye Adsorption

Clay-based materials also remove different dyes from water bodies to make them environmentally friendly. For the removal of dyes, clay and its modified forms are commonly used because of their low cost and eco-friendliness. Clay-based materials show better regeneration, adsorption capacity, and selectivity than other adsorbents due to their low cost, high porosity, and large surface area. Organobentonite, clay, montmorillonite, modified zeolite, modified pillared clay, and zeolite are used for the adsorption of different kinds of organic pollutants, such as chlorophenol, oil, BTEX, phenol, and methylene blue, with better adsorption capacities in the range of 60–90% [82,83]. The adsorption mechanism of dyes over clay-based materials involves the following steps. Initially, the diffusion of dyes occurs through layers of the materials, followed by diffusion through intraparticles, and finally, adsorption take place at the surface of clay-based materials [84,85]. External conditions such as pH, temperature, and solvent are highly responsible for the adsorption and restoration of dyes over clay materials. NaOH and HCl have been commonly used to restore adsorbents by changing the pH of the solution and subsequently withholding the charged state of adsorbents or adsorbates. Similarly, by enhancing the temperature from 30 to 50 °C, the adsorption efficiency was decreased to 50%. Solvent (acetone) is a better solvate for organic compounds, and has been employed to restore modified hydrotalcite for basic dye (safranin) adsorption from aqueous solutions [86]. Figure 5 [87] displays the adsorption of congo red dye over carbon-coated clay-based adsorbent. There are two possible adsorption mechanisms explained in Figure 5. The first possible mechanism is chemisorption between the congo red molecules and carbon-coated clay. This will occur between the Al atoms and O atoms of the SO_3_^−^ group. The second possible mechanism is physical adsorption via pore filling. However, of the above two adsorption mechanisms, the superior one is chemisorption, as above of 70% adsorption occurs via chemisorption [87].

The above study confirmed that chemisorption is mainly responsible for the dye adsorption of clay-based materials.

### 2.5. Layered Double Hydroxide-Based Materials for Dye Adsorption

Layered Double Hydroxides (LDH) were also used to remove and absorb dyes to improve the environment. LDH was first discovered in 1842. Due to its similarity with talc and the existence of a massive amount of water, it was named hydrotalcite [88]. LDHs comprise individual layers of brucite-like assemblies [Mg(OH)_2_]. All the layers in this construction are neutral electrically, with a cation of magnesium positioned in the center of an octahedron and six other hydroxyl groups located at the vertices. Positive charges are produced on the layers when Mg cations are substituted by trivalent metal cations, which stabilize by the presence of different anions in the interlayer space [89]. Different kinds of metal cations (divalent and trivalent) can be castoffs for the fabrication of LDHs, such as Mg, Co, Fe, Ni, Cu, Zn for M^2+^ and Ga, Fe, Al, Mn, and Cr for M^3+^; additionally, LDHs have the extraordinary capability to bind with various kinds of inorganic, organic and organometallic anions. Due to this diverse phenomenon, the fabrication of different types of LDHs with various physicochemical features is conceivable. Due to their flexible structure, distribution of high positive charge on their surfaces, and interlayer anion exchangeability, both kinds of LDHs (pristine LDHs and double-oxide LDOs) can be formed by the thermally treated method (calcination). These LDHs have multiple applications in the fields of catalysis, photochemistry, electrochemistry, pharmaceutics, and adsorption [90]. There are very few literature reports on the adsorption of dyes over LDHs. Khajeh et al. reported MB and MR adsorption studies over the Co/Fe–LDH@UiO-66-NH_2_ LDHs, and removal efficiency was 94.6–98.6% from tap water, river water, and groundwater [91]. Kostic et al. reported that the removal efficiency of Fe/Cu/Ni–LDHs for RB19 was 92.6–76.4% [92]. Prakash et al. reported biopolymer-modified LDH and LDHC extrudates in which LDHC extrudates showed better adsorption performance due to their high porosity [93]. The dyes are adsorbed over LDHs through electrostatic interaction because charges are present in the network of LDHs, which are mainly responsible for electrostatic interaction, as shown in Figure 6 [90]. The suggested mechanism comprised numerous interactions, such as the strong electrostatic interaction of LDHs with dye molecules, π-π interaction between the aromatic ring of dye molecules and LDHs, hydrogen bonding, and physical adsorption in the pores of LDHs [94]. The time taken for the adsorption of dyes and to reach adsorption equilibrium over different LEDs was in the range of 90–150 min, and the amount adsorbed was in the range of 10–150 mg/L, which is quite satisfactory and acceptable on the scientific level [95]. Table 2 displays the reported LDHs for various dye adsorptions.

### 2.6. MOF-Based Composite Materials for Dye Adsorption

MOFs are highly crystalline and porous 3D materials constructed from different ligands and metal ions via co-ordination bonds. MOFs are a highly porous framework of metal and organic fragments that integrate metal ions, metal oxides, hydroxides, and other ingredients into their network via doping, complex formation, and impregnation. MOFs are unique in different aspects: highly stable thermally and mechanically, light in weight, large surface area, and have the properties of tunable pores, diverse configuration and construction, and multiple metal sites. The above qualities make MOFs unique for different applications, e.g., adsorption, gas storage, catalysis, and biomedicine. Table 3 shows the building blocks of different popular and modified MOFs. Table 4 shows the different MOFs reported for the adsorption of various water-soluble dyes. The adsorption capacities of pure MOFs are not as good compared to modified MOFs, as already discussed in the literature. MOFs are used for the adsorption of methylene blue, congo red, rhodamine B, rhodamine 6G, methyl orange, etc.; for all these dyes, modified MOFs show better adsorption as compared to pure MOFs. Electrostatic interaction was followed by these adsorption studies depending on the nature of MOFs and dyes, as shown in Equation (8) and Equation (9), respectively.

Figure 7 [109] shows the adsorption mechanism of anionic dye (congo red) over cationic MOFs. The adsorption depends on the nature of the charge on the fabricated MOFs. If the MOFs are cationic, anionic dyes will adsorb efficiently on their surfaces and vice versa. MOFs have a positive charge on their network that easily attracts anionic dyes. Thus, such electrostatic interaction is dominant in MOFs for dye adsorption. π–π interaction, hydrogen bonding, ion exchange, Lewis acid–base interaction, and physical adsorption are other interactions responsible for dye adsorption over MOFs [121,122,123].
(8)$$Dye^{+}+MOF^{-}=MOF-Dye$$
(9)$$Dye^{-}+MOF^{+}=MOF-Dye$$

### 2.7. Biosorbents

Biological sources are used for multiple applications (dye adsorption, biofuel production, catalysis, etc.) due to their environmentally friendly nature [124,125]. Biosorbents mainly originated from biological sources and bio-organisms such as algae, fungi, bacteria, yeast, plants, and even the shells of animals [126,127,128,129]. Biosorbents have a strong affinity for the adsorption of various organic pollutants, which are discussed below in detail. Here, we will discuss chitin, alginate compounds, peat, biomass, and agricultural wastes as biosorbents for the adsorption of different dyes to make the environment clean and friendly for living organisms [129].

Among biosorbents, chitin is an essential natural biopolymer used abundantly after cellulose. It is a nitrogenous polysaccharide that is hard and inelastic in nature. Depending on the number of units of N-acetyl-glucosamine present in the biopolymer, it can be named chitin (units above 50%) and chitosan (units below 50%), respectively. Alginate is also a biopolymer and typically found in brown seeds, composed of linear polysaccharides and highly water soluble. Being water-soluble has limited its implementation for eradicating radionuclides and pollutants. They are used for applications after processing their ion exchange reaction with multivalent metal ions. After processing, they can be cast off as a bioadsorbent or for numerous pollutants. Peat is also a good biosorbent for various pollutants as it is relatively inexpensive, abundant, and readily available. Raw peat is composed of lignin, cellulose fulvic, and humic acid. It is a highly porous composite soil material with a massive amount of organic matter (humic ingredients) at various decomposition steps. The presence of humic acid and lignin can form a chemical bond with dye molecules. They also have different functional groups, such as aldehydes, ketones, carboxylic acids, phenolic hydroxides, alcohols, and ether. Peat is divided into four groups depending on the source materials: woody peat, moss peat, sedimentary peat, and herbaceous peat [126]. The efficiency of biosorbents for dye adsorption has been well demonstrated and explained by many researchers [130], as shown in Table 5. Living or dead biomass, such as fungi, white rot, and various microbial cultures, also have great potential for bioadsorption. During the fermentation process for microbial biomass production, many by-products are generated, which can be used as biosorbents for the pollutants. A large variety of fungi have the ability to decolorize various organic dyes. Different functional groups, such as thiol groups and phosphate amino carboxyl present in the fungus cell wall, facilitate the dye molecules binding with the fungi. Plant and agricultural waste (corn stalks, rice straws, wheat straws, sugarcane bagasse, etc.) are also valuable biosorbents, rich in cellulose with better hydrophilicity, and can be used as an alternative for dye adsorption from water bodies [130,131,132,133,134].

True and comprehensive biosorption mechanisms are still incompletely comprehended, but many researchers have tried their best to explain these mechanisms to some extent. They explained that two types of adsorption (chemisorption and physio-adsorption mechanisms) occur during this process. In chemisorption, mechanisms mainly include electrostatic interaction, ionic exchange, chemical precipitation, and complexes of functional groups (–RO, –COO, R–COOH, and –ROH) present on the biosorbent outer surface interacting with dye molecules and other organic pollutants. The physio-adsorption mechanism comprises the interaction between dye molecules and biosorbents through Van der Waals forces. It is stated that physical sorption has no vital and essential role in the biosorption mechanism, but few studies claim that physical sorption has a prominent role in adsorption [135,136]. Figure 8 [130] displays the proposed adsorption mechanism of crystal violet (CV) dyes over lemongrass leaf fibers incorporated with cellulose acetate (TLGL-CA). This mechanism indicated a strong adsorption of crystal violate on the TLGL-CA by electrostatic interaction between the negatively charged groups of TLGL-CA and positively charged quaternary ammonium groups in the CV in the basic medium [137]. Additionally, hydrogen bonding also formed between nitrogen atoms in CV and hydrogen atoms in hydroxyl groups of the TLGL-CA. Dipole forces could also be generated between partially negative charges on oxygen atoms in carbonyl groups of TLGL-CA and positive charges in CV [130]. From the above literature data, it is obvious that biosorbents are a useful candidate for environmental protection in the near future because of its highly friendly, abundant nature and cost-effectiveness.

**Table 5 molecules-28-01081-t005:** Adsorption of dyes over different biosorbents.

Biosorbent	Dye	Adsorption Capacity (mg/g)	Ref.
Bagasse	Acid blue 80	391 mg/g	[138]
Modified corn stalk	Methylene blue	328.46 mg/g	[131]
Coconut shell	Methylene blue	50.6	[139]
Straw waste	Rhodamine B	1.9	[139]
Wood powder	Methylene blue	850.9	[126]
Typha Latifolia	Methyl orange	50.34	[133]
Rice husk	Direct Red-31	74.074	[140]
Banana peel powder	Reactive Black 5	49.2	[127]
Hickory wood	Congo red	221.8	[141]
Lemongrass leaf	Crystal violet	36.10	[130]
Sunflower stem-pith	Methylene blue	346.32	[142]
Fish scale	Ponceau S	35	[129]
Loofah	Methylene blue	409.67	[143]
Moringa oleifera *Lam*. seeds	Acid Blue 9	329.5	[144]
Soybean husk	Basic Red 9	46.1	[145]

### 2.8. Factors Affecting Dye Adsorption

Adsorption is highly affected by numerous factors, such as dye concentration, amount of the materials used, contact time, surface area and pore volume of the materials, temperature, pH, and mode of treatment. Adsorption is highly dependent on pH. Cationic dyes are adsorbed more over the materials with negative charges on their surfaces, and anionic dyes are adsorbed more over the materials with more positive charges on their surfaces. Furthermore, cationic dye adsorbs more at a pH greater than pH_pzc_ (pH at the point of zero charges). In contrast, anionic dye adsorbs more at a pH less than pH_pzc_ when the surface is positively charged [146]. Dye concentration is also responsible for the level of adsorption of the materials. Adsorption becomes better as the concentration increases until the concentration is equal to the binding sites; subsequently, the adsorbed dye molecules initiate repulsion, and the adsorption capacity reduces [147].

Adsorption is a highly temperature-dependent process. The adsorption of dyes follows an exothermic or endothermic process. In the case of an endothermic process, the adsorption capacity enhances with the enhancement of temperature. This can be endorsed by the increased mobility of the dye molecules and functional adsorptive sites owing to the increasing temperature. On the other hand, in the case of an exothermic process, the adsorption decreases with an increase in temperature. An increase in temperature decreases the binding capability between the active sites on the adsorbent surface and dye molecules, resulting in reduced uptake of dye molecules [148]. The adsorption of dyes increases with the amount of material used because more sites will be available for the adsorption. However, the dye adsorbed at equilibrium decreased with higher adsorbent amounts. Contact time for the adsorption of dyes to attain an equilibrium over the materials is also significant to gain the highest adsorption efficiency in a short time. This factor is highly related to the material’s porosity [149]. If the material is porous, the diffusion will be fast, and the adsorption will reach its equilibrium in a short time compared to non-porous material, as discussed in detail in the above paragraphs. Agitation speed is also responsible for the increased adsorption of dyes because it facilitates the fast diffusion of molecules in dense materials and will ultimately increase the adsorption efficiency [150]. In contrast, a few materials are already highly porous, such as cryogels, in which there is no need for agitation because their absorption capacity is rapid at approximately 60–120 s, which is suitable for fast diffusion and more adsorption; from the above factors, it is quite clear that adsorption phenomena are highly responsible for different conditions [48]. The size of the dye molecule is also a prominent factor that affects the adsorption efficiency. Few dyes have small molecular structures and will be readily adsorbed in the pores of porous materials and vice versa.

## 3. Photocatalytic Degradation and Its Mechanisms

Adsorption is a fruitful route for removing dyes, but it cannot remove organic pollutants from water bodies altogether. Because the efficiency of designed systems for the adsorption is not high enough, the photocatalytic degradation route was implemented to degrade water-soluble organic dyes to make the environment suitable for aquatic organisms and human beings, keeping the above problem in mind. This is a prolonged and time-consuming process but still quite environmentally friendly and economical. Photocatalytic degradation can be divided into three types [151]:Charge-injection dye sensitization;Oxidation/reduction dye degradation (indirect degradation route);Photolysis of dye (direct degradation route).

In the first route, photons absorb equal energy or higher energy than the band gap of the respective materials, causing electron excitation from the lower band (valence band) to the higher band (conduction band) of the respective materials; as a result, the formation of pairs of electrons and positive holes occurs [152]. The generated holes and electrons mainly interact with dye molecules and produce excited dye molecules, denoted by dye*. In the next step, excited dye* (unstable dye) is converted to free radicals (anionic dyes (dye^−^) or cationic dyes (dye^+^)). In the last step, these generated dye radicals spontaneously degrade because free radicals are highly unstable and need stability, which is the main reason for dye degradation [153]. The above photocatalytic phenomenon is shown in Figure 9.

The second well-known mechanism is indirect, in which the photon absorption occurs of equal energy or higher in the energy of the material, which is responsible for exciting the electron from LOMO level to HOMO level, which leads to the generation of electron pairs as well as positive holes. As a result, molecular oxygen is reduced to superoxide radicals and hydroxide radicals generated by the reaction of holes and molecules of water. The formation of electron and hole pairs under visible light is mainly attributed to the presence of a photo-catalyst. Therefore, photo-catalysts have displayed excellent photocatalytic degradation efficiency for water-soluble organic dyes [154].

The direct degradation route (photolysis) is a very steady and slow process independent of catalytic materials. Photolysis is completed in the following steps:(10)$$Metal\;Oxide+Light\;\to \;h_{metal\;oxide}^{+}+e_{Metal\;oxide}^{-}$$
(11)$$h_{Metal\;oxide}^{+}+H_{2}O\;\to OH$$
(12)$$e_{Metal\;oxide}^{-}+O_{2}\to \;O_{2}^{-}$$
(13)$$O_{2}^{-}+H_{2}O\;\to OH$$
(14)Dye Molecule+OH →Degradation Products

For photo-degradation, using semiconductors is a good choice. Among the most photo-catalytic materials, TiO_2_ will be the main component because it is a better catalytic material than others. The following is the photocatalytic degradation of methylene blue dye over Ag_2_O composite material [155].

### 3.1. Materials Used for the Photocatalytic Degradation of Water-Soluble Organic Dyes

It is well-known that TiO_2_ (a semiconductor) has been used for decades as a photocatalytic material due to its cost-effectiveness, availability in the earth’s crust, and environmentally friendly nature. Good physical and chemical stability is also shown by TiO_2_ [156]. TiO_2_ has a band gap of 3.0 eV, which suggests that UV radiation shorter than 387 nm can be helpful for the photo-degradation of organic dyes [157]. However, the generated hole and electrons have a fast recombination ability to reduce the photocatalytic efficiency of such a catalyst. We also know that only 5% of UV radiation is available in the total solar spectrum, where the remaining 95% falls in the visible region. With the availability of more visible light, researchers have diverted their attention to designing a catalyst that shows more photocatalytic efficiency in the visible region. Due to the fast recombination rate of generated holes and electron pairs, pure metal oxides, such as TiO2, ZnS, ZnO, and WO2 are not efficient enough to accomplish much higher photocatalytic activities [158,159]. The band gaps of pure TiO_2_, transition-metal-doped TiO_2,_ and non-metal-doped TiO_2_ are shown in Figure 10 [160]. To overcome this problem, different doped and composite materials were introduced with better photocatalytic efficiency for dyes and other applications. Different materials, such as GO hybrid-metal-oxide-based photo-catalysts, MOF-based photo-catalysts, and LDH-based photo-catalysts are used as better photo-degradation materials for different kinds of water-soluble organic dyes, which are discussed one by one in detail.

### 3.2. Carbon Materials for Photocatalytic Degradation of Dyes

Carbon materials such as graphene oxide (GO) have the remarkable properties of electron transport and act as a better electron-accepting agent. A large number of oxygen moieties in GO make it an excellent choice to bind tightly with TiO_2_ and ZnO and voluntarily diffuse on the GO surface, which enhances the photocatalytic performance of GO. The band gap of GO can be modulated by varying the oxygen moieties on the GO surface. Partially oxidized GO acts as a semiconductor, while completely oxidized GO acts as an insulator. The band gap of GO is 3.26 eV, determined through UV-visible spectroscopy. GO possesses a distinctive electronic property, better mechanical strength, low density, better catalytic activity, large surface area and excellent electron-transport properties, contributing to decent absorptivity and spatial charge separation in hybrid structures.

Compared with GO, rGO has much more promising applications due to its higher electron mobility, better optical properties, chemical stability, high conductivity, high surface area, and higher thermal stability [161,162]. GO/rGo can make composites using different types of metal oxides/metals and act as a good photo-catalyst for harmful organic contaminants, and are usually cast off in fuel cells, such as Fe_2_O_3_/ZnO, La/TiO_2_, CuO/TiO_2_, BiOBr, Ag/ZnO, ZnFe_2_O_4_, Ag_3_PO_4_, Bi_2_Fe_4_O_9_, BaCrO_4_, CuFe_2_O_4_, W_18_O_49_, ZnO/ZnFe_2_O_4_, Cu_2_O, WO_3_, BiOI, Ag/Ag_3_PO_4_, BiVO_4_, ZnO, TiO_2_, Mn_2_O, Mn_3_O_4_, COFe_2_, Cu_2_O/SnO_2_, Bi_5_Nb_3_O_15_, ZnWO_4_, Nd/TiO_2_, SnO_2_, SnWO_4_, Bi_2_WO_6_, Ta_2_O_5_, ZnFe_2_O_4_, CdSe–TiO_2_, La2Ti_2_O_7_ [163,164,165,166,167,168,169]. The incorporation of GO/rGO in the above-mentioned metals/metal oxides reduces the band gap and can efficiently reduce the recombination of generated holes and electron pairs, making the photo-catalyst more effective for photo-degradation. GO has multiple functionalities which tightly bind with photocatalytic materials, such as ZnO, TiO_2_, etc., and diffuse on the GO surface, enhancing the photocatalytic efficiency of GO-based materials [170,171]. Figure 11, illustrates the radical generation through a semiconductor-doped GO or rGO binary composite, showing the valance and conduction bands. When light strikes the surface of the TiO_2_-GO photocatalyst, electrons and holes are generated, and the electrons in the conduction band of TiO_2_ are transferred to the surface of GO. This mechanism makes logic energetically, as the energy is somewhat higher than the TiO_2_ conduction band (−4.2 eV). The oxygen moieties in the GO sheet have p-electrons on the surface, enabling them to trap the coming electrons from TiO_2_. The present unpaired electrons in the oxygen atoms of GO can bind with the Ti atom for TiO_2_ to form the Ti-O-C bond and thus enhance the light-adsorbing range of TiO_2_. After absorbing the light, the radical formation reacts with the organic pollutants and the last degradation occurs [160].

The dye degradation occurs through the following mechanism, which is shown step by step using GO/RGO-doped semiconductors:(15)$$MO_{x}+hʋ\;\to \;MO_{x}\;\left[h^{+}+e^{-}\right]$$
(16)$$MO_{x}\;\left[h^{+}+e^{-}\right]+GO/rGO\to \;MO_{x}\;\left[h^{+}\right]+GO/rGO\;\left[e^{-}\right]$$
(17)$$GO/rGO\;\left[e^{-}\right]\;+\;H_{2}O_{2}\;\to \;OH^{●}\;+\;OH^{-}+GO/rGO$$
(18)$$MO_{x}\;\left[h^{+}\right]+OH^{●}+Dye\;\to Degradation\;Products$$

In the above equation, *MO*_*x*_ stands for metal oxide, *h*^+^ stands for the hole, *e*^−^ stands for electron, and OH^·^ stands for hydroxyl radical.

Numerous researchers have already reported designing different carbon-based composite materials with metal oxides for the photo-degradation of different organic pollutants. CeO_2_@activated carbon was reported by Hui Wang et al. with better photocatalytic degradation of acid orange 7 (AO7) dye in water media. Their photo-degradation efficiency was much better than other ceria-based composite materials [172]. Similarly, U. Bharagav et al. and Yuhan Li et al. reported bi-functional g-C_3_N_4_/carbon nanotubes/WO_3_ ternary nanohybrids materials and graphitic carbon nitride (g-C_3_N_4_) materials with better photocatalytic activities and for other environmental applications [167]. From the above literature, it is evident that carbon-based materials have a wide range of implementations in the photocatalytic degradation of different water-soluble organic dyes.

### 3.3. MOF Materials for Photocatalytic Degradation of Dyes

As discussed previously, MOFs are a miscellaneous class of three-dimensional co-ordination polymers fabricated using metal ions, and poly-functional organic ligands have a potential application in various fields [173,174]. In assessing this, d10 configuration-based MOFs have been discovered as visible and UV-active catalysts for various organic pollutants [175]. Tunable open metal center characteristics, ligands, and guest encapsulation in MOF pores are the main reason for providing numerous active sites for synergistic catalysis. Compared with other semiconductors, such as metal chalcogenides and metal oxides, MOFs have a highly available surface area, and the pore size and chemical environment of the MOFs can be easily modulated. The organic linkers present in the MOFs act as antenna chromophores upon light irradiation and characterized by orbital (delocalized) through conjugated Pi bond, which can be responsible for relocating the charges to metal-oxo cluster at the start of the photo-redox activity. Introducing a Fe-oxo cluster or NH_2_-based aromatic can shift the MOF’s response towards the visible light region. Encapsulation of metal nanoparticles (MNPs) in MOF pores acts as an electron trapping point and more active reaction site, which makes the lifetime of the charge-separated state suitably extended; this is mainly dangerous to photocatalytic activities but can be proficiently controlled. Considering the above positive aspects, MOFs can be implemented as an efficient candidate for photo-catalysis. The first MOF-based photo-catalyst (Zn_4_O(BDC)_6_ (MOF-5)) was reported in 2007 for the degradation of phenol [176]. Since 2007, increasing numbers of MOF-based photo-catalysts (Co, Ni, and Zn-MOFs) have been reported for the degradation of various dyes such as Rhodamine B, Brilliant Blue R, orange G, Remazol, and methylene blue [177]. Figure 12 displays the detailed mechanism of photo-degradation of Rhodamine dye using flower-like MOFs by Qin et al. [178]. The photo-degradation was completed using the following steps:(19)$$MOF+hʋ\;\to MOF\;\left[e^{+}+h^{+}\right]$$
(20)$$O_{2}+e^{-}\;\to \;˙O_{2}^{-}$$
(21)$$H_{2}O_{2}+e^{-}\;\to ˙OH+OH^{-}$$
(22)$$˙O_{2}^{-}+h^{+}\;˙OH+RhB\;\to Degradation\;Products$$

Numerous researchers have reported different MOF-based materials for the photo-degradation of dyes. Zhang et al. reported the photo-degradation of two MOF-based photo-catalysts ([Cu(4,4′-bipy)Cl]_n_ and [Co(4,4′-bipy)·(HCOO)_2_]_n_) for the degradation of methylene blue dye under a 500 W xenon lamp irradiation in the open air at room temperature. ([Cu(4,4′-bipy)Cl]_n_ and [Co(4,4′-bipy)·(HCOO)_2_]_n_) optical absorption band was 695 nm and 579 nm, from which the calculated band gap (E_g_) was 1.78 eV and 2.14 eV, respectively, via Tauc plot. The lower E_g_ values enhanced the light absorption and the material’s photo-catalytic efficiency [179]. Fe-based MOF/polymer composites were reported by Brahmi et al. as an efficient photocatalyst for the degradation of organic dyes. The photodegradation was completed (95%) in 30 min under UV-visible light irradiation [180]. Xiong et al. also reported Cd(II) MOFs for the photo-degradation of organic dyes. They explained that the band gap is significantly decreased in Cd(II) MOFs, which is the main reason it shows better photocatalytic activities. From the above observation, it is obvious that MOFs are a better candidate for the degradation of organic pollutants and to make the environment suitable for living beings. The mechanism of photocatalytic degradation of RhB via MOFs is shown in Figure 11 [178].

### 3.4. LDH Materials for Photocatalytic Degradation of Dyes

Layered double hydroxides (LDHs) and LDH-based materials are promising photocatalysts for the photodegradation of different kinds of organic pollutants. The mechanism of photodegradation of different organic dyes via LDHs has previously been explained in the literature with detail [154,181,182]. LDHs have excellent photocatalytic properties because of their tunable band structure in the range of 2.2 to 3.2 eV and remarkable photo-responsive nature. LDHs have highly cost-effective and near use for practical applications. Recently, different kinds of complex LDHs (from simple bi-cationic MgAl-LDHs to tri-cationic or tetra-cationic multi-phased species such as MgFeTi-LDHs, MgZnAlFe-LDHs, post-calcination multiphase metal oxide LDHs) have been reported, all of which have shown promising and improved photocatalytic performance [182,183,184]. In the literature, several studies on LDHs and LDH-based photo-catalysts have been reported for the photo-degradation of azo-based dyes. For example, ZnAl–LDHs loaded with metal oxides, doped with rare-earth metals, or mixed metal oxides are the most popular photo-catalysts for photo-degradation studies. Zhang et al. studied the photocatalytic effect of ZnAl–LDO complexes using Orange II dye as a model reaction, and the photo-degradation efficiency was 74.3% in 100 min, which was much better than commercially available TiO2 and ZnO [185]. In another study, Morimoto et al. reported a series of ZnAl–LDHs and MgAl–LDHs for the photo-degradation of methylene orange (MO) and fast green (FG) with better photocatalytic performance [186]. Rhodamine B was also photodegraded by CdS/CoAl–LDHs as reported by Cai et al. The photocatalytic degradation of Rhodamine B was completed at 72% in 30 min and 100% in 60 min at room temperature, which was entirely satisfactory in comparison with other reported materials [187]. Due to the complex structure of dye molecules, the performance of the photocatalytic material, mechanisms, kinetic study, free radical generation, reactivity, and the whole pathway require more systematic studies and advanced techniques. For comprehensive mechanistic identifications, fundamental advances in investigation with interdisciplinary determinations are required [181]. Figure 13, displays the general mechanism of photocatalytic degradation of dye over the pristine and modified LDHs. Kumar et al. reported a CuF-based LDH which was used as a photo-catalyst for the photo-degradation of methylene blue with 41.56% efficiency [188]. By doping reduced graphene oxide (RGO) and zirconium (Zr) over the CuF-LDH, the photodegradation efficiency was enhanced to 95.2 and 91.5%, respectively. Electrons were elevated from the valance band (VB) and promoted into the conduction band (CB) of the LDHs composite catalyst; simultaneously, a hole was produced in the VB [189]. As a result, molecular oxygen was reduced to superoxide radicals and hydroxyl radicals generated by the reaction of the hole and water molecules. The formation of electron and hole pairs under the visible light was mainly attributed to the presence of CuO and RGO. In this mechanism, CuO performs its responsibility as an electron donor, and RGO performs its responsibility as an electron acceptor. Electrons migrated from the CuO and Fe_2_O_3_ CB to the RGO CB in ZrRGOCuFe-LDH, owing to low band energy level. The electrons produced by RGO mainly interact with O2 to generate superoxide radicals, whereas holes interact with water or OH to generate OH radicals. Additionally, Cu(I) could alter Fe(III) to Fe(II) in CuFe-LDH or ZrRGOCuFe-LDH. Therefore, ZrRGOCuFe-LDH displayed tremendous photocatalytic degradation efficiency for methylene blue dye [188].

### 3.5. Factors Affecting Photocatalytic Degradation

Photocatalytic degradation of dyes is dependent on various factors, such as dye concentration, catalyst dosage, pH effect, temperature, size and structure of photocatalytic material, surface area, nature of the pollutants, light intensity and radiation time, as well as the effect of the dopant [190]. The dye concentration is an essential factor that will decide the efficiency of the designed photocatalytic materials. Typically, the photocatalytic degradation of dyes decreases with an increase in dye concentration and vice versa. With an increase in dye concentration, more dye molecules will adsorb on the surface of the photocatalyst, and as a result, fewer generated photons will be available to reach the surface of the photocatalyst. As a result, less OH^●^ will be formed, which will be the leading cause of reduced photo-degradation of organic pollutants [191]. The amount of catalyst also affects photocatalytic degradation prominently. With the increase in the dosage of photocatalysts, the active site number will increase, and as a result, more OH^●^ radicals will be generated, which take part in the photocatalytic degradation of dyes. However, beyond a specific point, the solution will become highly turbid, affecting the reach of UV radiation towards the whole system and decreasing the photodegradation efficiency [192]. Material porosity is a vital factor that will decide photocatalytic efficiency. If the material is porous, it will be easy for dye molecules to interact with the active site of the photocatalyst, resulting in degradation. Among the above-discussed materials, cryogels and MOFs are highly porous materials and will be the better candidates for photocatalytic degradation of different kinds of organic pollutants [193]. Temperature is another common factor that is highly responsive to enhancing or decreasing the reaction kinetic. In photocatalytic studies, the photocatalytic degradation was enhanced with increased temperature, as reported by Ahmed et al. [194]. The effect of temperature was studied at 25 °C to 85 °C. The photocatalytic degradation was enhanced up to 65 °C, but after 65 °C, the photo-catalyst efficiency was decreased. From the results, it is clear that 91.28% of Alizarin Yellow R (AYR) dye was degraded when the temperature reached 65 °C in 42 h. This whole process confirms that this reaction is endothermic. The increase in temperature follows the kinetic molecular theory that with an increase in temperature, the mobility of the dye molecules becomes faster and has a chance of more interaction with light radiation [195]. Above 65 °C, the efficiency was low and might be accredited to the weakening of the adsorptive forces between the dye molecules and the active site of the photocatalytic material. Time duration also has a pronounced effect on the photocatalytic degradation of dyes. AYR was studied for up to 42 h under photoradiation using an FeNPs photo-catalyst. The degradation was fast in the initial 6 h, reaching 58.71%. The degradation further increased and reached 92.51% with an increase in time (42 h). These data confirm that photocatalytic degradation increases with an increase in time duration [194]. The efficiency of photocatalytic degradation also affects the presence of ionic species and other organic compounds in the dye solution. The presence of carbonate, bicarbonate, chloride, nitrite, nitrate, and phosphate ions highly affects the generation of photo-electrons, recombination of electron-hole, and scavenging of hydroxyl radicals. Among the above radicals, chloride has the most detrimental effect on photocatalytic degradation, which is scavenging for holes and hydroxyl radicals [196]. Solution pH is also highly responsible for the photocatalytic degradation of dyes. The change in pH is responsible for the surface charge variation on the photo-catalyst, which will alter the potential of the photocatalytic reaction. The photocatalytic degradation is also affected by the nature of the dyes (anionic and cationic). In other words, the anionic dyes act like a strong Lewis base and have a chance to adsorb easily on the surface of positively charged photocatalysts. This favors the dye adsorption under an acidic environment; however, in a basic environment, such a complexation route is unlikely to be favored because of the competition of hydroxyl group adsorption and molecules of dye, in addition to the Columbic repulsion due to the negatively charged photo-catalyst with the dye molecule [197]. The pH mainly defines the surface charge of the photocatalyst. Minimum adsorption of dyes occurs when the solution pH is at the isoelectric point (point of zero charge). The photo-catalyst surface is charged positively below the isoelectric point and carries a negative charge above it [198].

## 4. Chemical Degradation Approaches for Water-Soluble Dyes and Nitro-Compounds

Photocatalytic degradation is a good method for degrading different dyes, but this route is prolonged and time-consuming. Furthermore, photocatalytic degradation is not useful for the nitro-compounds that are more stable than dye molecules. With this in mind, we will next focus on the chemical degradation of different water-soluble dyes and nitro-compounds using different catalysts in the presence of a reducing agent. Different chemical approaches were implemented for the chemical degradation of water-soluble dyes, such as degradation of dyes using a reducing agent (chemical method), electrochemical method, coagulation–flocculation, and advanced oxidation approaches. Among the approaches mentioned above, the electrochemical degradation route is more expensive than other approaches for dye degradation [199]. The main reason for reduced implementation of the electrochemical approach is high energy consumption, the requirement for more chemicals, and costly equipment [200,201]. Other challenges in implementing the electrochemical approach are the production of metabolites and other by-products during these processes. Here, we will mainly focus on the chemical degradation route of different dyes.

### 4.1. Chemical Degradation of Azo, Triarylmethane, and Anthraquinone Dyes

Several industries are highly responsible for disposing of different water-soluble dyes in different water bodies, which is an alarming situation for aquatic organisms and human beings. To ensure the safety of aquatic organisms and their effects on human beings, researchers introduced different catalytic materials using reducing agents (NaBH4, H_2_O_2_, LiBH_4_, Ca(BH4)_2_, Mg(BH_4_)_2_, LiAlH_4_ and ammonia boranes, etc.) and different catalytic materials for the degradation of different water-soluble dyes. Every water-soluble dye has a proper chromophore, giving each dye a unique color. The dyes can be degraded by reducing their chromophore through a proper mechanism using different catalysts [202,203]. The azo-based dyes are quite different when using enzymatic degradation, because in enzymatic degradation, the N=N group is directly responsible for the degradation, but using other chemical degradation routes is quite different. The azo-based dyes have different side functional groups, which highly affect the degradation efficiency. The degradation of the dye reactive red 120 is more rapid than reactive orange 16 dye. The main reason for fast degradation is a more reactive site in the reactive red 120, such as a sulfone compound which will attract the radical ions more efficiently than the orange 16 dye. The number of azo bonds is also an important parameter for degradation. Fewer azo bonds (Methylene orange) will degrade rapidly compared to more azo bonds (Congo red) and vice versa. The second point of note during the degradation of azo dyes is the breakdown of the C-N bond rather than N=N, as in enzymatic degradation, although the latter is quite probable to occur. The most vital aspect is that a similar mechanism is followed even though the dyes belong to unlike groups. The breakdown of the C-N bond is quite spontaneous and effortless, with the formation of no complex structure. Typically, after the breakdown, deamination and desulfonation occur afterward, the final products of naphthalene and benzene are formed due to the chemical breakdown of the dyes. Overall, we conclude that azo-based dyes decolorize easily compared to others [204]. The structures of several azo dyes are shown in Figure 14 [205].

The degradation of other dyes is different from azo-based dyes, and the azo group is not responsible for the degradation, but the chromophore, rather than the azo bond, is mainly responsible for the degradation. In some dyes, such as triaryl methane-based dyes, the central carbon bridge (triple benzene/benzene ring derivative) is responsible for the dye degradation. The degradation of another famous and common dye, anthraquinone, is also due to the breakdown of chromophores rather than azo bonds inside the complex structure of the dye. The possible degradation mechanism may be related to the cross breakdown and series breakdown. Cross breakdown will lead to the formation of benzoic acid as a final product, while in a series breakdown, the end product will depend on the presence of other atoms and compounds in the benzene ring. In dyes such as anthraquinone, the breakdown will lead to benzene and phthalic acid forming as a final product.

### 4.2. Chemical Degradation of Nitro-Compounds and Mechanisms

Nitro-compounds have many applications in different fields of life, but their hazardous effect is very prominent, as with nitrobenzene (NB), an essential intermediate of nitro-aromatic compounds with high toxicity. 2-nitrophenol (2-NP) and 4-nitrophenol (4-NP), also considered primary pollutants owing to their high solubility and stability in water, have become part of a vital environmental debate and have gained significant attention in recent years. Different approaches have been used to eliminate the nitro-compounds, but the catalytic reduction is the better route for degradation because nitro-compounds are highly stable [23,206]. The presence of the nitro group in nitrobenzene with solid electron-withdrawing capability decreases the electronic cloud density, making the NB difficult to degrade easily [207]. Similarly, nitrophenol also harms the environment and living organisms because it is primarily used in the dye, pesticide, and plasticizer industries [208]. Researchers have used different materials to degrade nitro-compounds and convert them into environmentally friendly ones. Using a reducing agent, NB can convert into less hazardous compounds such as nitrosobenzene (NOB) phenylhydroxylamine and aniline via chemical degradation [209]. The reducing agent (NaBH_4_ and ammonia borane) is dissolved in the water and produces a hydrogen source, which is highly essential for the reduction reaction as shown in the following equations (Equation (23) and Equation (24), respectively) [210]:(23)$$NaBH_{4}+2H_{2}O\;\to NaBO_{2}+4H_{2}$$
(24)$$NH_{3}BH_{3}+2H_{2}O\;\to NH_{4}^{+}+BO_{2}^{-}+3H_{2}$$

Similarly, dangerous NP can also degrade into an environmentally friendly compound, aminophenol, using different catalytic materials in the presence of a reducing agent [211], and the mechanism of degradation is displayed in Figure 15 [212]. The reduction of nitro-compounds depends on the nature of the designed system (catalyst), which is explained in Section 4.3 in detail. During this process in the first step, the catalyst converts the reducing agent (NaBH_4_, etc.) to molecular H_2_, along with BO^2-^ generation, which is dissociated on the catalyst surface. In the second step, the adsorption of nitro-compounds (4-NP, 2-NP, NB, etc.) occurs on the catalyst surface such that the interaction with these dissociated H_2_ species and the reduction happens in a stepwise manner to convert nitro-compounds to amino compounds. The resulting amino compounds are desorbed from the catalyst surface and the catalyst becomes ready for the subsequent cycles [23].

The kinetics of such reduction reaction of dyes and nitro-compounds always follow the pseudo-first-order kinetic model because the reducing agent used during this reaction is always in access. The apparent rate constant value (*K_app_*) can be calculated from the slope by plotting ln(C_t_/C_0_) against time. Initially, C_0_ is noted (absorbance value at zero time of dyes/nitro-compounds), followed by the C_t_ value with time intervals (1 min, 2 min, 3 min, etc.). In the next step, the C_t_ values are divided by C_0_ and take the *ln* of all the values. In the last step, the value of ln(C_t_/C_0_) is plotted on the Y-axis and time on the X-axis, and from the slope, one can obtain the apparent rate constant value (*K_app_*). From the *K_app_* value, activation energy (*Ea*) can be calculated using the Arrhenius equation (Equation (25)), which can tell whether the reaction is kinetically feasible. From the *Ea,* one can calculate entropy and enthalpy by applying the Eyring equation (Equation (26)).
(25)lnK=lnA−Ea/RT
(26)$$ln(K/T)=∆H/RT+ln\left(\frac{k_{b}}{h}\right)+∆S/R$$

### 4.3. Materials Used for the Chemical Degradation of Dyes and Nitro-Compounds

For the chemical degradation of dyes and nitro-compounds, mostly noble metal nanoparticles are incorporated in 3D cross-linked polymeric materials or other porous composite materials to reduce their aggregation ability and make them a valuable candidate for chemical degradation. Among the noble metal nanoparticles, Iron (Fe), cobalt (Co), nickel (Ni), copper (Cu), gold (Au), silver (Ag), palladium (Pd), Platinum (Pt), and so on, are the most commonly used nanoparticles for the degradation of various dyes and nitro-compounds [213,214,215]. Among the above nanoparticles, nickel and cobalt are the most unstable nanoparticles and become rapidly oxidized when exposed to air. However, they have an excellent degradation and reduction ability for different kind of dyes and nitro-compounds. The other nanoparticles, such as silver, gold, palladium and iron, are the most stable nanoparticles and can be used as catalysts for a long time without any fear of oxidation if exposed to air. The nanoparticles are primarily loaded in different kinds of templates, such as polymers (microgels fabricated via emulsion polymerization, hydrogels fabricated via conventional free radical polymerization, and cryogels fabricated via cryo-polymerization) [216,217,218], and MOFs [219,220]. In the next step, the nanoparticles mentioned above are loaded in the templates via the reduction route using the salts of the respective nanoparticles along with the reducing agents [221,222] or through the thermal reduction method [38,42]. After the reduction, each kind of nanoparticle gains its specific color, which indicates the successful fabrication of hybrid materials. Silver nanoparticles have a dark brown color, gold nanoparticles have a raspberry color and the remaining nanoparticles have a dark black color as an indication. During chemical degradation, specific reducing agents (NaBH_4_, H_2_O_2_, citrate, ethylene glycol, etc.) are used, depending on the nature of the catalyst. Reducing agents alone cannot degrade or reduce the dyes or nitro-compounds. The degradation and reduction mechanisms are shown in Figure 15 [212]. Porosity is also the main parameter that is highly responsible for the fast degradation and reduction of dyes and nitro-compounds, because diffusion occurs during such a reaction and the reactant molecules strike active sites of the catalyst as a result, converted into a friendly product. If the diffusion is not rapid and fast, the degradation and reduction will take a long time, and the catalyst cannot be considered a good candidate for the catalysis. For example, hybrid cryogels (fabricated at cryogenic conditions) and conventional hybrid hydrogels (fabricated at room temperature) are entirely different in their catalytic behaviors due to their pore structure and morphology, as previously discussed in the literature in detail [43]. Conventional hydrogels have tiny pores and closed cavities in comparison with cryogels. The difference is mainly due to synthetic procedures and conditions, although free radical polymerization is followed in both [223]. The main difference is the cryogenic condition in which one can observe two regions on the microscopic level. The first region is called “poly-crystal”, which is vital for hetero-porosity (different pore size), and the second region is called “unfrozen liquid microphase” (UFLM), which is a more vital and essential region for cryo-polymerization [224,225]. Both the regions are highly affected by negative temperature (temperature below the freezing point of solvent). MOF is also a porous material, as explained well in Section 2.6. There is not much literature on hybrid MOFs, but in recent years, few articles have been available on the catalytic degradation of dyes and nitro-compounds. Khosravi et al. reported palladium nanoparticles incorporated with MOFs that act as porous and efficient catalytic materials for dye degradation, such as Rhodamine B and methyl red in the presence of a reducing agent [219]. Similarly, platinum-loaded MOFs were used to reduce Nitrophenol into aminophenol with better catalytic performance and almost wholly reduced to 8 min [220]. NiFe_2_O_4_ nanoparticle-loaded MOFs were also used to reduce nitrobenzene as an efficient porous catalyst [226]. These literature studies confirm that MOFs are also a better candidate to act as a stable template for nanoparticles inside their 3D network.

### 4.4. Factors Affecting Chemical Degradation

Multiple factors are responsible for the chemical degradation of dyes and nitro-compounds. The main factors are pH, temperature, porosity, the catalyst’s nature, concentration, the dye’s nature, template functionality, dose rate, etc. All of these factors are well explained in multiple research articles and review articles [227,228]. If the template has a pH-sensitive functional group like COOH-, then with an increase in pH, negative charges will increase throughout the template which will repel each other and porosity will enhance; as a result, diffusion of dye molecules and nitro-compounds will rapidly strike the nanoparticles and will convert into product, and vice versa. Temperature is also a prominent factor affecting the catalyst’s performance [218]. At higher temperatures, molecules’ motion becomes faster according to kinetic theory and will strike the catalyst rapidly to convert into products and vice versa [38,42]. Porosity is an important factor that highly affects the performance of a catalyst. For example, hybrid hydrogels have a dense structure and closed cavity, and are not considered the best catalyst compared to hybrid cryogels with a macroporous structure, and diffusion of reactant molecules will be rapid. Overall, hybrid cryogels are better catalytic materials due to their cryogenic synthesis route. The other factors also affect the catalyst performance for dye and nitro-compound degradation [229]. After factors, advantages and disadvantages of the three mentioned methods are shown in Table 6.

### 4.5. Reusability of the Above Materials

Reusability is an essential parameter of any designed system. In our study, all the materials can be reused for a few cycles, but their removal efficiency decreases with each cycle. Among them, polymeric materials and carbon-based sponges have better reusability than the other materials. MOFs, LDHs, and clay-based materials can be reused for the subsequent cycles by centrifugation, but during centrifugation there is a chance that they will lose some amount of the absorbent, which will affect the efficiency of the materials for the next cycle. Overall, the materials mentioned above have shown better adsorption performance after a few cycles.

### 4.6. Economic Cost

The cost of adsorbing and catalytic materials is a crucial factor that should be considered for practical applications. However, only few literature reports are available on the cost estimation of these fabricated materials, as most of the research is limited to the laboratory scale, which the researchers always neglect. According to the literature, the average cost of activated carbon was one dollar for 2.0–2.2 kg [236]. However, it is difficult to measure the cost of those fabricated materials; it is worthless to ignore the properties and efficiency of adsorbents. Hence, cost estimation of absorbents and catalytic materials is a comprehensive process with numerous factors that should be considered, such as transportation, synthetic route, availability, and lifetime issues [237]. In general, these materials’ modification methods (drying, autoclaving and crosslinking) enhance their cost. Aside from this, the cost of transportation and regeneration should not be neglected. To our knowledge, few researchers pay attention to the cost evaluation of novel adsorbing and catalytic materials. Thus, an in-depth study is needed to investigate this comprehensive area [34] further.

## 5. Conclusions and Future Prospects

This review article highlights three aspects of removing and degrading different water-soluble organic dyes. The stability of dyes, different dyes, and the sources responsible for the disposal and pollution of water bodies were also explained in detail. Among the dyes, removal and degradation aspects were adsorption, photocatalytic degradation, and chemical degradation using a reducing agent. In terms of the adsorption of dyes, the interaction mechanism of dyes over the specific materials, different formulas for the kinetic calculation, and materials used for dye adsorption were discussed in detail. The second part of this review article concerns photocatalytic degradation. The mechanism of photocatalytic degradation, the effect of dopant on the efficiency of photocatalytic degradation, and different catalytic materials for photocatalytic degradation implemented for this purpose were the main aims of this review article. The third and last part of this review article briefly covers the chemical degradation of dyes and nitro-compounds over different catalytic materials and which material was the better candidate for the degradation. In each dye, the chromophore is the leading functional group responsible for giving a specific color to each dye. Here, it is also explained that some dyes have single chromophores and a few have multiple chromophores, and degradation of multiple chromophores is complex compared to a single functional group. The key finding of this review article was the effect of side functional groups on the stability of chromophore and the final product after dye degradation. Different catalytic materials were also discussed and used for the chemical degradation of organic dyes. Nitro-compounds were also briefly highlighted in this review article. With increasing growth and scientific advancement, synthetic materials are expected to solve most environmental problems in the near future. There is a significant and vital need to develop new advanced water technologies to improve drinking water quality, eliminate micro-pollutants, and strengthen industrial fabrication methods by using flexibly adaptable water treatment structures. Nanomaterials, particularly synthetic sorbents and catalysts, deliver the potential for developing water technologies that can be easily pragmatic for customer-specific applications. This review article will be beneficial for readers working in this field. Because there are many review articles published on such kinds of studies, we covered all three aspects together, making it easy for readers to read one review article and understand the three phenomena.

## Figures and Tables

**Figure 1 molecules-28-01081-f001:**
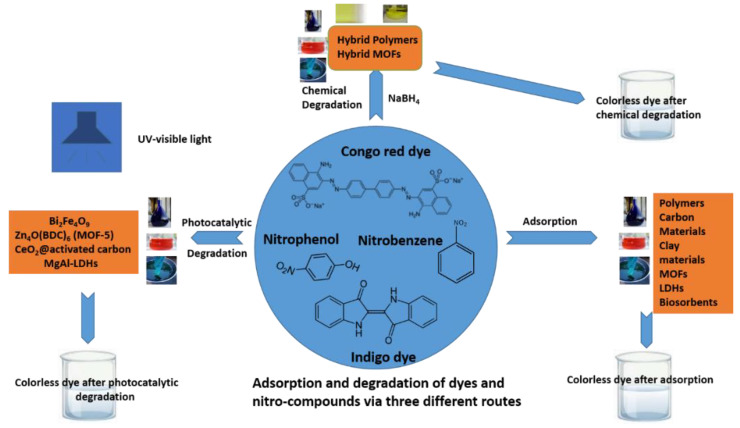
Graphical representation of adsorption, photocatalytic degradation, and chemical degradation of dyes and nitro-compounds using different materials.

**Figure 2 molecules-28-01081-f002:**
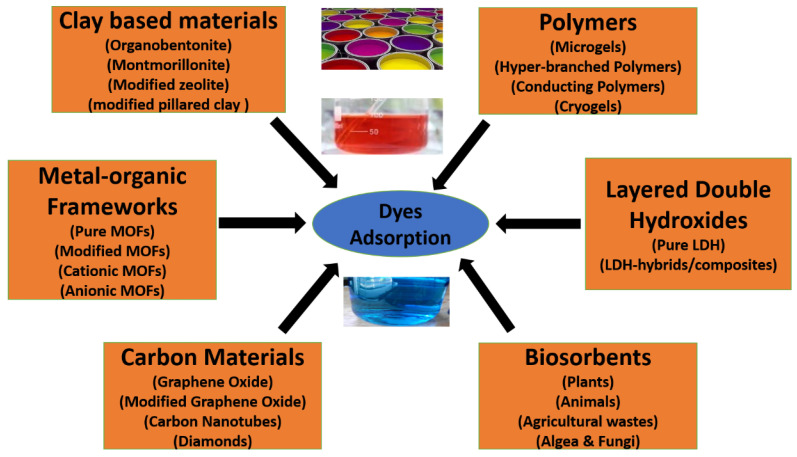
Dye adsorption over different kinds of designed materials.

**Figure 3 molecules-28-01081-f003:**
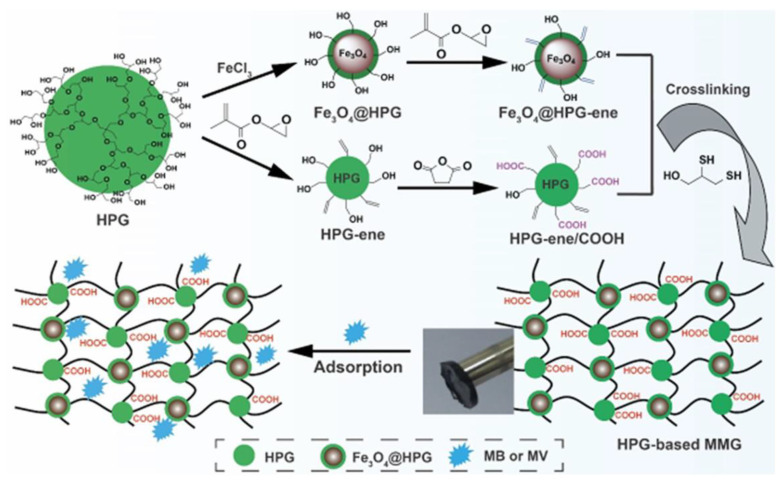
Magnetic hyperbranched polymer for dye adsorption via different interactions. Reprinted from ref. [2] with permission.

**Figure 4 molecules-28-01081-f004:**
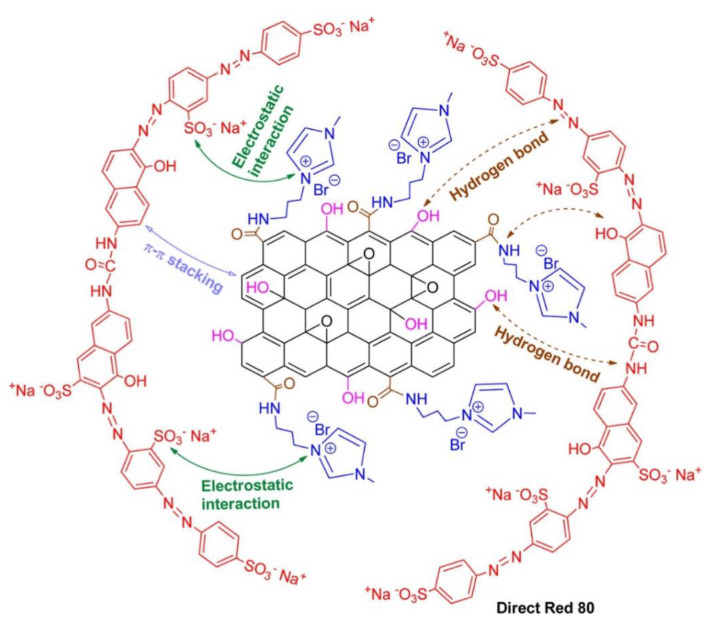
Adsorption of dyes over carbon-based materials via different interactions. Reprinted from ref. [34] with permission.

**Figure 5 molecules-28-01081-f005:**
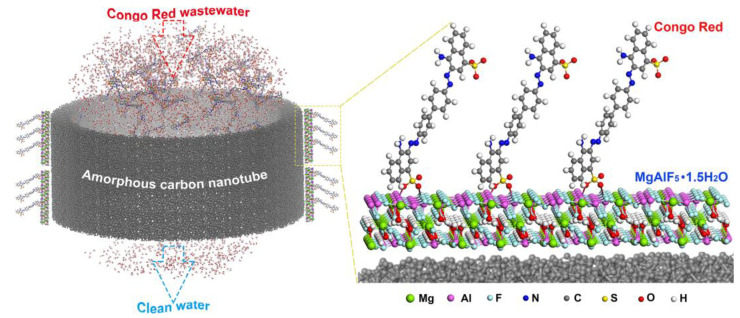
Proposed mechanism of dye adsorption over carbon-coated clay. Reprinted from ref. [87] with permission.

**Figure 6 molecules-28-01081-f006:**
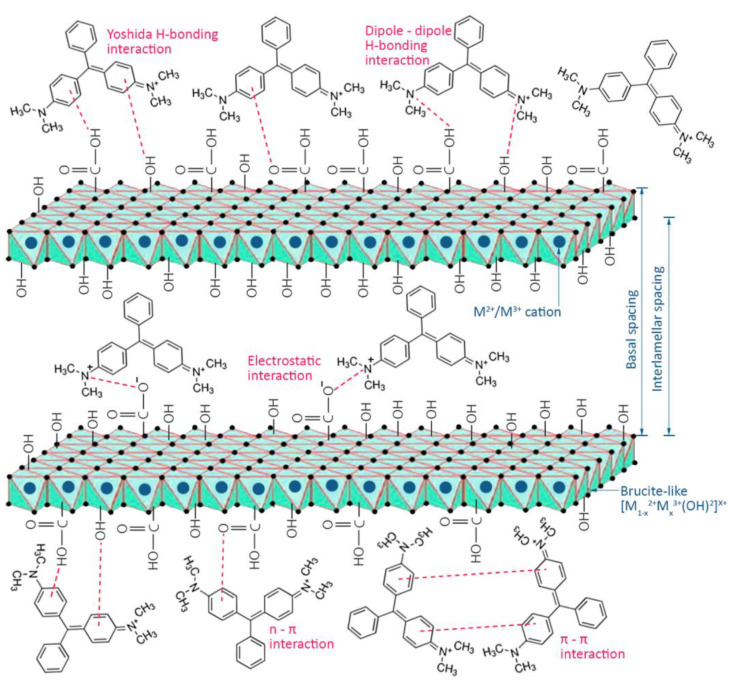
Schematic diagram of possible interactions in LDH with dye molecules. Reprinted from ref. [90] with permission.

**Figure 7 molecules-28-01081-f007:**
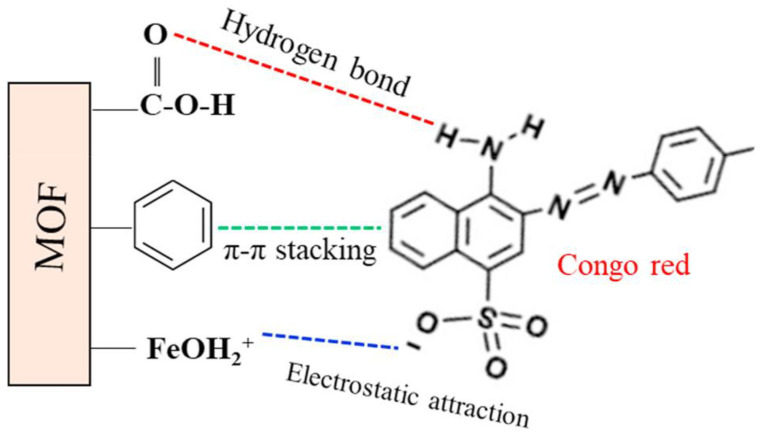
Various interactions of dye molecules during adsorption over MOFs. Reprinted from ref. [108] with permission.

**Figure 8 molecules-28-01081-f008:**
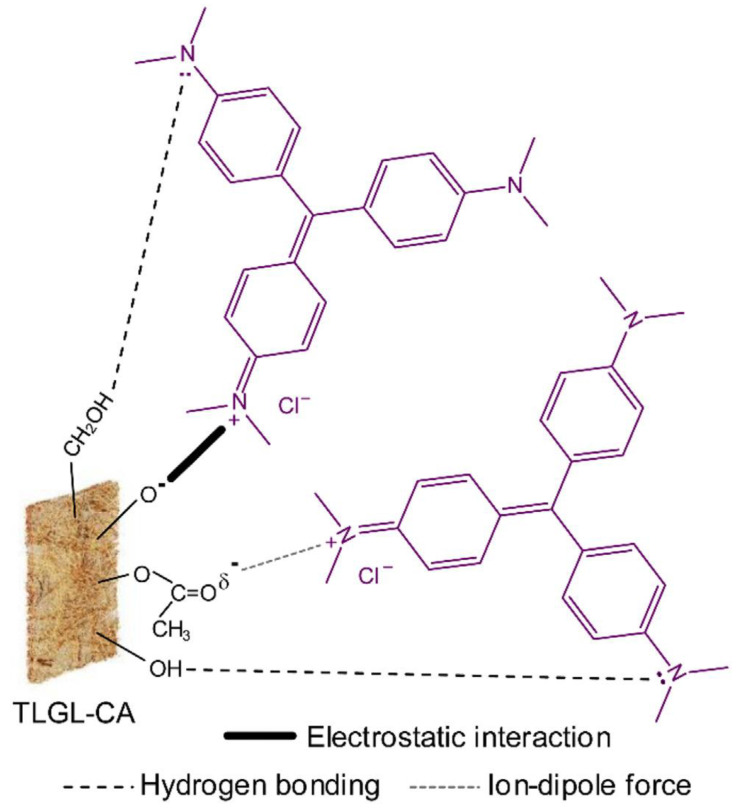
Proposed adsorption mechanism of crystal violet on lemongrass leaf fibers incorporated with cellulose acetate. Reprinted from ref. [130] with permission.

**Figure 9 molecules-28-01081-f009:**
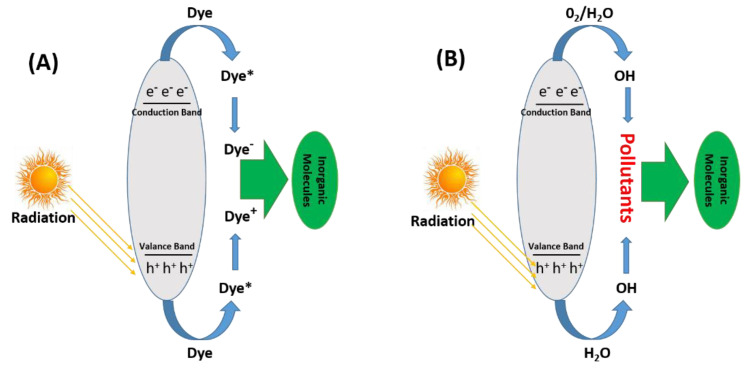
Dye photo-degradation through sensitization route (**A**); mechanism of photocatalytic treatment of pollutants (**B**) (Dye* for unstable dye).

**Figure 10 molecules-28-01081-f010:**
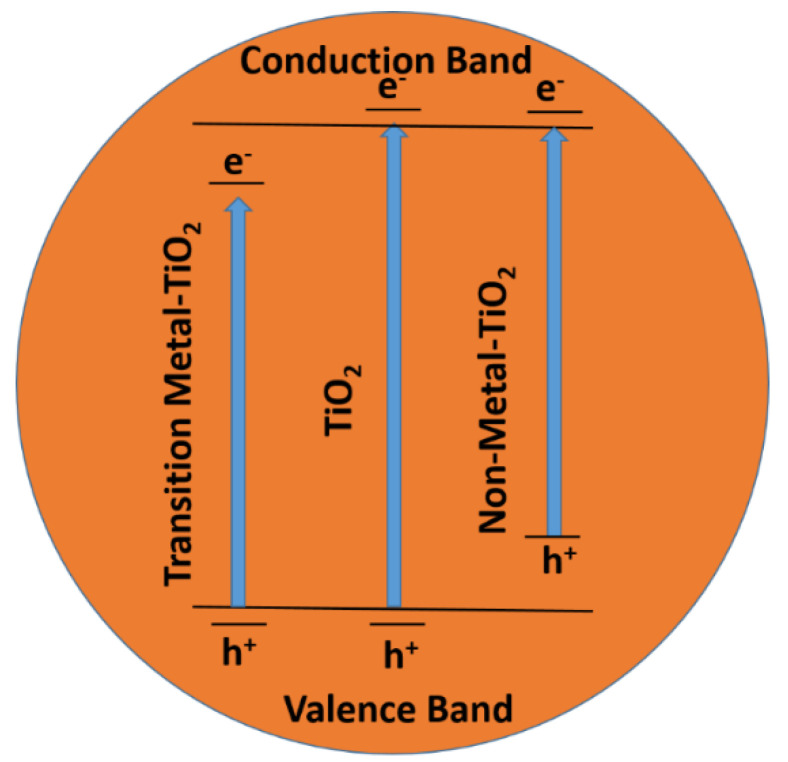
The energies of the valence band levels and conduction band level energies of transition-metal-doped TiO_2_, pure TiO_2_, and non-metal-doped TiO_2_.

**Figure 11 molecules-28-01081-f011:**
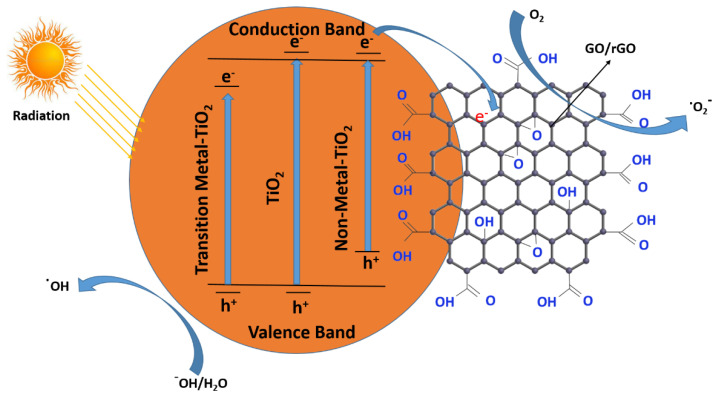
A sketch of the radical generation through semiconductor-doped GO or rGO binary composite, showing the valance band and conduction band.

**Figure 12 molecules-28-01081-f012:**
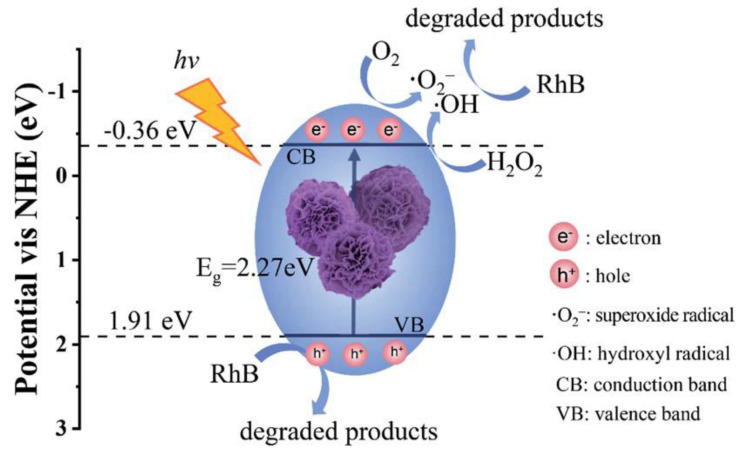
The mechanism for photocatalytic degradation of RhB via MOFs. Reprinted from ref. [178] with permission.

**Figure 13 molecules-28-01081-f013:**
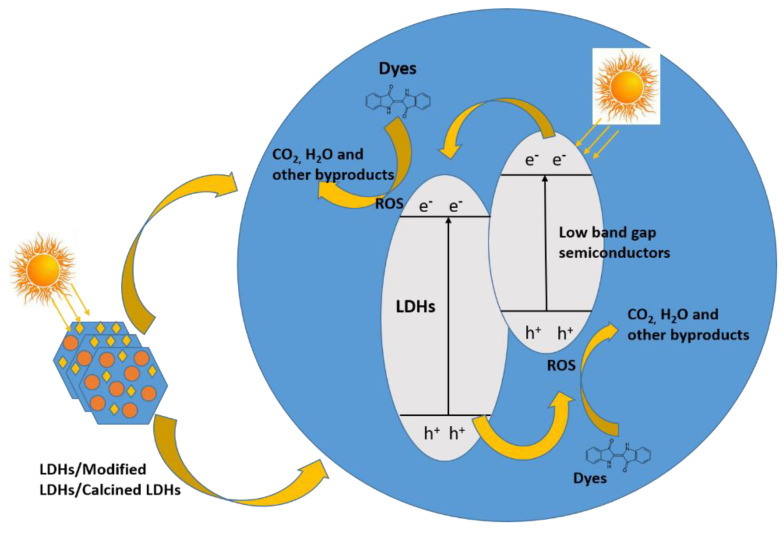
Schematic representation of photocatalytic degradation of dyes over LDH-based composite materials.

**Figure 14 molecules-28-01081-f014:**
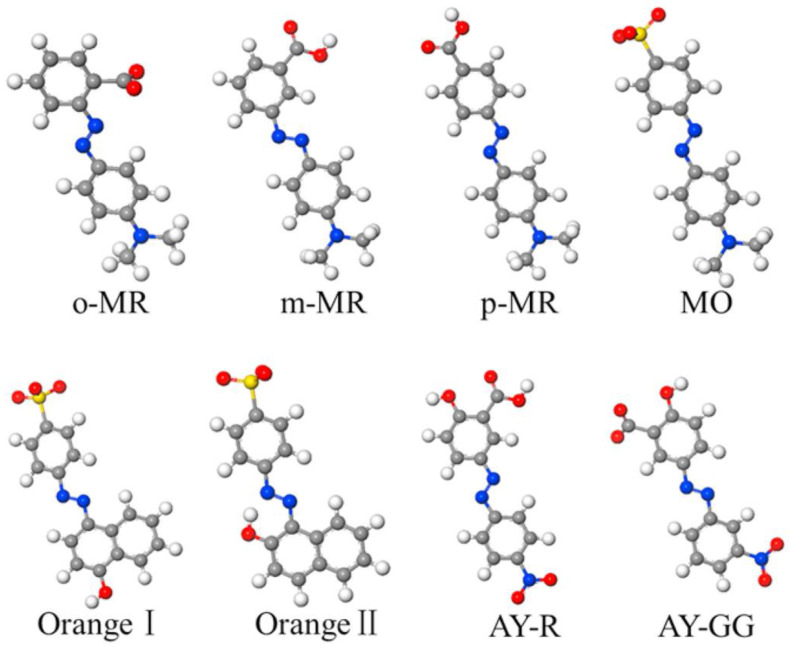
Molecular structure of several azo dyes (ortho-methyl red, meta-methyl red, para-methyl red, methyl orange, methyl orange I, methyl orange II, alizarin yellow R, and alizarin yellow GG). The gray, red, blue, yellow, and white balls represent carbon, oxygen, nitrogen, sulfur, and hydrogen atoms, respectively. Reprinted from ref. [205] with permission.

**Figure 15 molecules-28-01081-f015:**
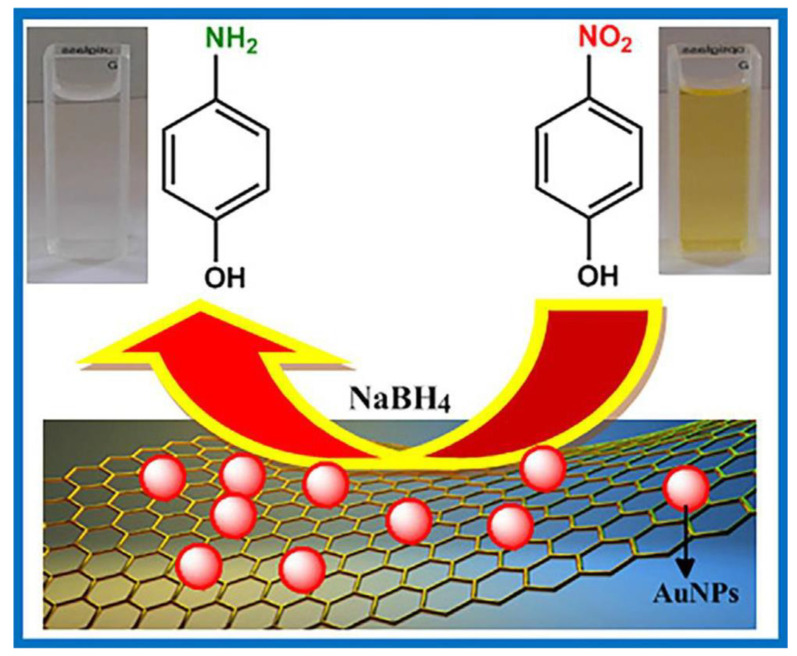
The reduction mechanism of p-Nitrophenol using a reducing agent and catalyst. Reprinted from ref. [212] with permission.

**Table 1 molecules-28-01081-t001:** Description of different types of dyes.

Type	Description	Applicability
Reactive dyes	Dyes which have a chromophore capable of making a strong covalent connection on the fabric’s nucleophilic sites.	Silk, wool, cotton, lenin, rayon
Dispersive dyes	Less soluble dyes in water cause dispersion and give color to fabrics via H-bonding and other Van der Waals forces.	Acrylic, polyester, nylon
Direct dyes	Attach to fabrics via adsorption isotherm. For the adsorption of dyes, electrolytes of an inorganic base are added, which highly promotes the uptake of dyes.	Linin, silk, rayon, cotton, wool
Acidic dyes	The dyes are mainly applied in acidic conditions (pH 2–6). Coloring occurs via ionic bonding between dyes and fabrics.	Wool, silk, nylon
Basic dyes	These dyes contain an amine group and have better solubility in alcohol. Coloring occurs via ionic bonding between the dye and negative charges on the network of fabrics.	Rayon, acrylic (solubility is limited)

**Table 2 molecules-28-01081-t002:** Previously reported LDHs for different dye adsorptions.

LHDs	Dye	Adsorption Capacity (mg/g)	Ref.
Cal. Mg/Al–LDH	Amaranth	0.704	[96]
Co/Zn/Al–LDH	Methylene blue	169.49	[97]
ZIF-67@Co/Al–LDH	Green bezanyl-F2B	57.24	[98]
Cal. Zn/(Al + Fe)–LDH	Methylene blue	487.9	[99]
ZIF-67@Co/Al–LDH	Methyl orange	180.5	[98]
DPA–Mg/Al–LDH	Eriochrome black-T	242.98	[100]
Mg/Al–Cl–Biochar –LDH	Methylene blue	406.47	[101]
WFS Mg/Fe–LDH	Congo red	9127.08	[102]
PVDF@Mg/Al–LDH	Methyl orange	621.17	[103]
NH_2_–Mg/Al–LDH–EDTA	Congo red	632.9	[104]
Co/Fe–LDH	Malachite Green	555.62	[91]
3D–Mg/Al–LDH	Congo red	1428.6	[105]
ZnO/SiO_2_–Zn/Al–LDH	Methyl orange	222.25	[106]
O3D–Mg/Al–LDH	Rhodamine B	377.89	[107]

**Table 3 molecules-28-01081-t003:** Building blocks of different popular MOFs and modified MOFs [108].

Popular MOFs	Modified MOFs	Modified MOFs
MIL-100 (Fe)	Ce(III)-doped UiO-67	UiO-66-NH_2_
UiO-66 (Zr)	H_6_P_2_W_18_@Cu_3_(BTC)_2_	UiO-66-P
Ni@MOF-74(Ni)	32%NT/MIL-100(Fe)	UiO-66/PGP
Cu-BTC	Fe(II)@MIL-100(Fe)	In-MOF@GO
ZIF-8	Fe_3_O_4_/MIL-101(Cr)	GO-Cu-MOF
MIL-125(Ti)	MIL-68(In)-NH_2_	Fe_3_O_4_/MIL-100(Fe)
In-MOF	MgFe_2_O_4_@MOF	
MIL-53(Al)	POM@UiO-66	

**Table 4 molecules-28-01081-t004:** Different MOFs were reported for the adsorption of various water-soluble dyes.

MOFs	Dye	Adsorption Capacity mg/g)	Ref.
Uio-66	Methylene blue	91	[109]
MIL-53(Al)–NH_2_	Methylene blue	45.97	[110]
MIL-68(Al)	Methylene blue	73.8	[111]
Fe_3_O_4_@MIL-100(Fe)	Methylene blue	1666	[112]
Fe_3_O_4_/Cu_3_(BTC)_2_	Methylene blue	84–245	[113]
PED-MIL-101	Methyl orange	194	[114]
In-MOF@GO	Rhodamine B	267	[115]
MIL-101(Cr)	Xylenol orange	322–326	[116]
Mn-MOF	Crystal violet	938	[117]
ZIF-67	Acid orange 7	738	[118]
MIL-53(Al)–NH_2_	Malachite Green	38.09	[110]
[Ni_2_F_2_(4,4′ bipy)_2_(H_2_O)_2_](VO_3_)_2_⋅8H_2_O	Congo red	242.1	[119]
TMU-8	Reactive black 5	79.39	[120]
MIL-68(Al)	Rhodamine B	1111	[112]

**Table 6 molecules-28-01081-t006:** Advantages and Disadvantages of Adsorption, Photo-, and Chemical Degradation.

Advantages	Disadvantages	Ref.
**Adsorption:** Dyes are removed through various sorbents such as polymeric materials, carbon-based materials, clay-based materials, LDHs, and MOFs. The above materials have low capital cost, high adsorption performance, and are flexible. The materials mentioned above can remove many dyes from water bodies. The reusability of sponge-like materials is also great compared to powder-like materials.	These materials do not degrade organic pollutants. They also require constant maintenance since saturated adsorbents dramatically reduce the adsorption performance. Carbon-based materials have high costs, although their adsorption capacity is much better.	[230,231,232]
**Photocatalytic degradation:** This is an environmentally friendly method of dye degradation because no extra energy is needed for it, fewer or no chemicals are needed for this process, and because solar energy is abundant in nature and we can efficiently utilize it for photocatalytic degradation. It is also a cost-effective route for dye degradation. The materials mentioned above for photocatalytic degradation can degrade many dyes from water bodies. The reusability of sponge-like materials is also great compared to powder-like materials.	Photocatalytic degradation is a time-consuming method in the modern world. Some materials such as LDHs and MOFs have costly synthetic routes such as hydrothermal methods, which require much energy. The reusability of the LDHs and MOFs is also an issue because they require centrifugation. During centrifugation, there is a chance that some amount of the materials will be lost.	[233,234,235]
**Chemical degradation** Photocatalytic degradation is a slow degradation process. On the other hand, chemical degradation is a fast degradation process for dye and nitro-compound degradation. These methods allow high concentrations of dyes and nitro-compounds to degrade quickly in the presence of reducing agents. Such work can be performed anywhere because it does not need an expensive instrument. For confirmation, only a UV-spectrophotometer is needed.	It is not an environmentally friendly process due to the use of reducing agents, and is a costly method compared to photocatalytic degradation.	[38,42,222]

## Data Availability

Not applicable.

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
