# Peer review of "A Comprehensive Review on Adsorption, Photocatalytic and Chemical Degradation of Dyes and Nitro-Compounds over Different Kinds of Porous and Composite Materials"

_molecules, 2023, doi:10.3390/molecules28031081_

Round 1

Reviewer 1 Report

Haleem et al. reported a "Comprehensive review on dyes adsorption as well as photocatalytic and chemical degradation of dyes and nitro-compounds over different kind of catalytic materials". This review article covers three aspects of removing organic pollutants from water bodies and highlights various materials reported for this purpose. Different hazards of dyes were explained comprehensively, which will be very useful for the environmental studies student. The authors also mentioned all the equations and kinetics in all three parts about adsorption, photocatalytic degradation, and chemical degradation in detail. The present work will be an excellent contribution to the field of environmental protection. The review article is publishable in molecules after addressing the following minor comments and suggestions.

Comments and suggestions

1.     The title needs to be revised and "Comprehensive review on adsorption, photocatalytic and chemical degradation of dyes and nitro-compounds over different porous and composite materials.

2.     The abstract need to be improved and comprehensive.

3. The introduction needs to add more information about nitro compounds and their adverse environmental effects.

4.     In section 2.2, more updated references about different kinds of polymers for adsorption should be added.

5.     In section 2.2, add more information about the cryogels that why it's preferred over conventional hydrogels and microgels.

6.     Section 2.3, also adds literature on carbon nanotubes for dye adsorption.

7.     Section 2.7, factor affecting dye adsorption, need improvement with more detail.

8.     Section 2.5, factors affecting photocatalytic degradation, need more improvement.  

9.     Degradation of the nitro-compounds section should be improved with more information about nitro-compounds and their chemical degradation.  

10. Please check the spelling mistake throughout the manuscript.

11. English need to be improved through English expert.

12. Conclusion needs to be comprehensive with future prospective.

13. The references style must be according to the Journal format.

Author Response

Reviewer 1:

Haleem et al. reportedComprehensive review on dyes adsorption as well as photocatalytic and chemical degradation of dyes and nitro-compounds over different kind of catalytic materials”. This review article is covering three different aspects to remove the organic pollutants from water bodies and also highlighted various materials that reported so far for this purposes. Different hazardous of dyes were explained in comprehensive way which will be very useful for the student of environmental studies.  The authors also mentioned all the equations and kinetics in all three parts about adsorption, photocatalytic degradation and chemical degradation in very detail manner. The present work will be a good contribution to the field of environmental protection. The review article is publishable in molecules after addressing the following minor comments and suggestions.

Comments and suggestions

  1. The title need to be revised and should be “Comprehensive review on adsorption, photocatalytic and chemical degradation of dyes and nitro-compounds over different kind of porous and composite materials.

Response: Thank you very much for such a positive suggestion. The title become change in the revised manuscript.

  1. The abstract need to be improved and comprehensive.

Response: Thank you very much for such a positive suggestion. The abstract become improved the revised version.

  1. In the introduction part need to add more information about nitro-compounds and it adverse effect on environment.

Response: Thank you very much for such a positive suggestion. Revised according to the reviewer suggestion.

  1. In section 2.2, more updated references about different kind of polymers for adsorption should be added.

Response: Thank you very much for such a positive suggestion. Revised according to the reviewer suggestion

  1. In section 2.2, add more information about the cryogels that why its prefer over conventional hydrogels and microgels.

Response: Thank you very much for such a positive suggestion. Revised according to the reviewer suggestion

  1. Section 2.3, also add literature carbon nanotubes for dyes adsorption.

Response: Thank you very much for such a positive suggestion. Revised according to the reviewer suggestion

  1. Section 2.7, factor effecting dyes adsorption need improvement with more detail.

Response: Thank you very much for such a positive suggestion. Revised according to the reviewer suggestion

  1. Section 2.5, factors effecting photocatalytic degradation need more improvement.
  2. Response: Thank you very much for such a positive suggestion. Revised according to the reviewer suggestion.
  3. Degradation of nitro-compounds section should be improved with more information about nitro-compounds and its chemical degradation.
  4. Response: Thank you very much for such a positive suggestion. Revised according to the reviewer suggestion.
  5. Please check the spelling mistake throughout the manuscript.
  6. Response: Thank you very much for such a positive suggestion. Revised according to the reviewer suggestion.
  7. English need to be improved through English expert.
  8. Response: Thank you very much for such a positive suggestion. We have improved the English throughout the manuscript.
  9. Conclusion need to be comprehensive with future prospective.

Response: Thank you very much for such a positive suggestion. Revised according to the reviewer suggestion.

  1. The references style must be according to the Journal format.

Response: Thank you very much for such a positive suggestion. Revised according to the reviewer suggestion.

Reviewer 2 Report

The manuscript entitled “Comprehensive review on dyes adsorption as well as photo-catalytic and chemical degradation of dyes and nitro-compounds over different kind of catalytic materials” provides some good results, but it has a lot of issues that must be resolved. Therefore, the current manuscript could be considered for publication, but after going through a major revision.      

1.      Avoid using keywords that are the same as the titles words as much as possible.   

2.      What are the maximum permissible limits of dyes and nitro-compounds? They should be discussed and compared to the concentrations being actually discharged.

3.      The lifecycle of dyes and nitro-compounds in different ecosystems and the bio-magnification of these pollutants should be discussed.

4.      The governmental rules, regulations, and policies imposed across the world to control the release of these toxic compounds should be discussed.

5.      The negative effects of dyes and nitro-compounds on the environment, organisms, and human health should be further elaborated (eg: negative impacts on the endocrine and reproductive systems in both aquatic organisms and humans) in a more organized way in the introduction section.

6.      Biological materials or commonly known as biosorbents should also be included in this review and discussed.

7.      Biochar should be discussed and included in the carbon material adsorbents.

8.      Fenton mechanism should be included in this review article as it is considered one of the most important degradation mechanisms of dyes and other toxic compounds.

9.      A detailed discussion should be provided on the processes taking place in figure 3.

10.  A detailed discussion should be provided on the processes taking place in figures 4 and 5.

11.  “Due to their flexible structure, distribution of high positive charge on its surfaces and interlayer anions exchangeability, both kinds of LDHs (Pristine LDHs and double oxide LDHs) can formed by thermal treated method (calcination).” double oxide LDHs must be called Layered double oxides (LDOs).

12.  Quality of figure 6 must be improved and a detailed explanation of the processes mentioned in this figure must be provided.

13.  Table 2 cannot be accepted because the references are missing and also there should be more experimental information provided in this table including specific pollutants that were removed and their concentration, the dose of LDHs, pH level, and the adsorption efficiency.

14.  The same corrections must be applied for table 3.

15.  More detailed information must be provided for the processes occurring in figure 7.

16.  The regeneration or recycling studies must be covered in this review to demonstrate the efficiency of the investigated adsorbents after being reused for several times.

17.  “Photo-initiated holes and electrons undergoes through multiple steps and finally generate OH radicals which is strong non-selective oxidizing agent and mainly responsible for the degradation of different kind of water soluble dyes and leads to finalize the mineralization” This is not an accurate description of the photocatalysis mechanism!! An accurate description must include the role of holes and hydroxyl radicals at the valence band (VB) as well as the role of superoxide radicals at the conduction band (CB). Also, the photo-generation of these radicals must be discussed in details.

18.   Quality of figure 10 should be improved.

19.  A detailed description must be provided for the processes occurring in figure 12.

20.  Quality of figure 12 should be improved.

21.  Quality of figure 13 must be improved.   

22.  A detailed description must be provided for the processes occurring in figure 14 and the quality of this must be improved.

23.  The economic factor should be discussed in this review by clarifying the production cost of the investigated materials and removal techniques and also indicating which one of the three treatment techniques is the best one based on this factor.         

24.  A detailed comparison between the three investigated techniques in this study must be drawn to clarify their advantages and disadvantages.

25.  The authors must provide more discussion and comparisons with literature throughout the whole manuscript. Some examples of the recent articles that could be useful for drawing such comparisons and enriching the manuscript are:

https://doi.org/10.1007/s11356-022-21871-x

https://doi.org/10.1021/acs.inorgchem.9b01862  

https://doi.org/10.1016/j.jenvman.2022.115238

https://doi.org/10.1007/s11696-019-00982-9

26.  The English language must be improved.  

Author Response

Reviewer 2:

The manuscript entitled “Comprehensive review on dyes adsorption as well as photo-catalytic and chemical degradation of dyes and nitro-compounds over different kind of catalytic materials” provides some good results, but it has a lot of issues that must be resolved. Therefore, the current manuscript could be considered for publication, but after going through a major revision.   

Response: Thank you so much for your precious time to review our review articles and give a positive suggestion to improve the manuscript further. We have tried our best to addressed all the issues raised by the reviewer 2 but few points were not necessary to include in this review article.   

  1. Avoid using keywords that are the same as the titles words as much as possible.

Response:   Thank you so much for your positive comment. We have done this change according the reviewer suggestion.

  1. What are the maximum permissible limits of dyes and nitro-compounds? They should be discussed and compared to the concentrations being actually discharged.

Response: Thank you so much for your positive comment. Such a permissible limit is not mentioned in the literature so far but how much organic pollutants are entering into the water bodies annually are mentioned in the introduction part.

  1. The lifecycle of dyes and nitro-compounds in different ecosystems and the bio-magnification of these pollutants should be discussed.

Response: Thank you so much for your positive comment. The dyes and nitro-compound is highly stable compounds that we have already discussed in the first draft of manuscript. Due to their stable structure a lot of materials designed to remove and degrade these organic pollutants from water bodies.

  1. The governmental rules, regulations, and policies imposed across the world to control the release of these toxic compounds should be discussed.

Response: Thank you so much for your positive suggestion. I have search such type of studies a lot but not find any related literature about it. Hope I will include such type of studies in the next review article which is about bisorbents.

  1. The negative effects of dyes and nitro-compounds on the environment, organisms, and human health should be further elaborated (eg: negative impacts on the endocrine and reproductive systems in both aquatic organisms and humans) in a more organized way in the introduction section.

Response: Thank you so much for your positive comment. The negative impact of dyes and nitro-compound on aquatic organism and human health is discuss with more detail in the revised version.

  1. Biological materials or commonly known as biosorbents should also be included in this review and discussed.

Response: Thank you so much for your positive suggestion. Biosorbents section is added in the revised manuscript.

  1. Biochar should be discussed and included in the carbon material adsorbents.

Response: Thank you so much for your positive suggestion. Added biochar according to the suggestion of the reviewer.

  1. Fenton mechanism should be included in this review article as it is considered one of the most important degradation mechanisms of dyes and other toxic compounds.

Response: Thank you so much for your positive suggestion. Here in this review articles we are only concentrate on the three mentioned methods for dyes adsorption and degradation. Fenton mechanism is little different than other three methods and not included in the present review article.

  1. A detailed discussion should be provided on the processes taking place in figure 3.

Response: Thank you so much for your positive comments. We have explained the possible interaction occurs in this figure and the whole figure is about click chemistry which is not needed here.

  1. A detailed discussion should be provided on the processes taking place in figures 4 and 5.

Response: Thank you so much for your positive suggestion. Explain with detail according to the suggestion of the reviewer. Figure 5 is change with better one in the revised manuscript.

  1. “Due to their flexible structure, distribution of high positive charge on its surfaces and interlayer anions exchangeability, both kinds of LDHs (Pristine LDHs and double oxide LDHs) can formed by thermal treated method (calcination).” double oxide LDHs must be called Layered double oxides (LDOs).

Response: Thank you so much for your positive comments. We have make this correction in the revised version.

  1. Quality of figure 6 must be improved and a detailed explanation of the processes mentioned in this figure must be provided.

Response: Thank you so much for your positive suggestion. Explain with detail according to the suggestion of the reviewer. The Figure is looking clear in the revised manuscript.

  1. Table 2 cannot be accepted because the references are missing and also there should be more experimental information provided in this table including specific pollutants that were removed and their concentration, the dose of LDHs, pH level, and the adsorption efficiency.

Response: Thank you so much for your positive comments. We have added a new table with references and deleted the old one.

  1. The same corrections must be applied for table 3.

Response: Thank you so much for your positive comments. We have added a new table 4 with references and the old table 3 description become change. The table 3 is about the building block of different popular MOFs and modified MOFs.

  1. More detailed information must be provided for the processes occurring in figure 7.

Response: Thank you so much for your positive suggestion. Explain with detail according to the suggestion of the reviewer.

  1. The regeneration or recycling studies must be covered in this review to demonstrate the efficiency of the investigated adsorbents after being reused for several times.

Response: Thank you so much for your positive suggestion. Explain with detail according to the suggestion of the reviewer in the revised manuscript.

  1. “Photo-initiated holes and electrons undergoes through multiple steps and finally generate OH radicals which is strong non-selective oxidizing agent and mainly responsible for the degradation of different kind of water soluble dyes and leads to finalize the mineralization” This is not an accurate description of the photocatalysis mechanism!! An accurate description must include the role of holes and hydroxyl radicals at the valence band (VB) as well as the role of superoxide radicals at the conduction band (CB). Also, the photo-generation of these radicals must be discussed in details.

Response: Thank you so much for your positive suggestion. The mentioned part is improved according to the reviewer suggestion.

  1.  Quality of figure 10 should be improved.

Response: Thank you so much for your positive suggestion. The figure magnification is enhanced in the revised manuscript.

  1. A detailed description must be provided for the processes occurring in figure 12.

Response: Thank you so much for your positive suggestion. New figure is added in the revised manuscript with detail description.

  1. Quality of figure 12 should be improved.

Response: Thank you so much for your positive suggestion. The new figure is added in the revised manuscript.

  1. Quality of figure 13 must be improved.  

Response: Thank you so much for your positive suggestion. New figure is added in the revised manuscript with detail description.

  1. A detailed description must be provided for the processes occurring in figure 14 and the quality of this must be improved.

Response:  Thank you so much for your positive suggestion. New figure is added in the revised manuscript with detail description.

  1. The economic factor should be discussed in this review by clarifying the production cost of the investigated materials and removal techniques and also indicating which one of the three treatment techniques is the best one based on this factor.  

Response:  Thank you so much for your positive suggestion. This part is added (section 2.6) in the revised manuscript and explain with detail according to the suggestion of the reviewer. Also explain the advantages and disadvantages of each method in the revised manuscript.  

  1. A detailed comparison between the three investigated techniques in this study must be drawn to clarify their advantages and disadvantages.

Response: Thank you so much for your positive suggestion. We have added the separate section (2.5) of advantage and disadvantages of all the three methods.

  1. The authors must provide more discussion and comparisons with literature throughout the whole manuscript. Some examples of the recent articles that could be useful for drawing such comparisons and enriching the manuscript are:

https://doi.org/10.1007/s11356-022-21871-x

https://doi.org/10.1021/acs.inorgchem.9b01862  

https://doi.org/10.1016/j.jenvman.2022.115238

https://doi.org/10.1007/s11696-019-00982-9  

Response:  Thank you so much for your positive suggestion. We have added the above mentioned references in the revised manuscript.

  1. The English language must be improved.

Response:  Thank you so much for your positive suggestion. We have improved the English of whole manuscript.

Reviewer 3 Report

The present manuscript described the the adverse effect of dyes and nitro-compounds. The adsorption of dyes in various materials have been deeply revealed to elaborate the adsorption process and models. Moreover, photocatalytic degradation and chemical degradation of dyes and nitro-compounds was also discussed in this manuscript. This paper can be recommended to be published. The authors are suggested to consider the following remarks and further improve the manuscript before submitting the final version.

1. Would the adsorption rate higher than the degradation rate under the low concentration of dyes ?

2. In the line of 723, the sentence of “ The degradation 722 and reduction mechanism is shown in the following Figure.”There is no figure followed

Author Response

Reviewer 3:

The present manuscript described the  adverse effect of dyes and nitro-compounds. The adsorption of dyes in various materials have been deeply revealed to elaborate the adsorption process and models. Moreover, photocatalytic degradation and chemical degradation of dyes and nitro-compounds was also discussed in this manuscript. This paper can be recommended to be published. The authors are suggested to consider the following remarks and further improve the manuscript before submitting the final version.

  1. Would the adsorption rate higher than the degradation rate under the low concentration of dyes?

Response: If the concentration of the dyes is very low the adsorption will be also low but the degradation process will be fast because it’s a chemical process using reducing agent. But degradation also depends on various factors as already discussed in the main parts.

  1. In the line of 723, the sentence of “The degradation 722 and reduction mechanism is shown in the following Figure.”There is no figure followed .

Response:  Thank you so much for your positive suggestion. We have correct this mistake in the revised manuscript.

Reviewer 4 Report

In this review article, the adverse effect of dyes and nitro-compounds on aquatic organism and human being were discussed in depth. The structure of dyes and its stability were also the main aim of this study. Types of dyes were also highlighted in introduction part of this review articles.

After assessing the whole manuscript, some issues should be solved.

(1)   In figure 2, a table should be added to compare the advantages and disadvantages of different adsorption materials.

(2)  The adsorption mechanisms of dyes on adsorbent should be summarized. And how these adsorption mechanism were confirmed.

(3)  A table should be added to compare the advantages and disadvantages of different photocatalytic material.

(4)  The outlook of this review should be underlined.

(5)  Some quantitative data should be added in the abstract and conclusions to highlight the main findings of this review.

Author Response

Reviewer 4:

In this review article, the adverse effect of dyes and nitro-compounds on aquatic organism and human being were discussed in depth. The structure of dyes and its stability were also the main aim of this study. Types of dyes were also highlighted in introduction part of this review articles.

After assessing the whole manuscript, some issues should be solved. 

  • In figure 2, a table should be added to compare the advantages and disadvantages of different adsorption materials.

Response: Thank you so much for your positive suggestion. We have added the separate section (2.5) of advantage and disadvantages of all the three methods.

  • The adsorption mechanisms of dyes on adsorbent should be summarized. And how these adsorption mechanisms were confirmed.

Response: Thank you so much for your positive suggestion. The adsorption mechanism was confirming by the color change and can also confirm by the spectra through UV visible spectrophotometer. FTIR and XPS can also use to confirm the adsorption of dyes over different sorbents. This is already mentioned in the manuscript.

  • A table should be added to compare the advantages and disadvantages of different photocatalytic material.

Response: Thank you so much for your positive suggestion. We have added the separate section (2.5) of advantage and disadvantages of all the three methods.

  • The outlook of this review should be underlined.

Response: Thank you so much for your positive suggestion. The outlook is explained in the conclusion and future prospective.

(5)  Some quantitative data should be added in the abstract and conclusions to highlight the main findings of this review.

Response: Thank you so much for your positive suggestion. But this is review article and not need of putting quantities date in the abstract and conclusion. All the quantities data is mentioned in the main part of review article.

Round 2

Reviewer 2 Report

The manuscript entitled “Comprehensive review on dyes adsorption as well as photo-catalytic and chemical degradation of dyes and nitro-compounds over different kind of catalytic materials” could now be accepted for publication, but there are two problems that must be fixed before that.      

1.       “As a result, molecular oxygen was reduced to superoxide radicals as well as hydroxide radicals generated by the reaction of hole and molecules of water.” It must be the hydroxyl radicals not the hydroxide radicals on page 24 and also it must be corrected in Figure 13.

2.      In tables 2, 4, and 5, there is no removal efficiency. In fact, what is mentioned and measured in mg/g is called adsorption capacity.   

Author Response

Thank you so much for your kind suggestions and comments. We have revised the manuscript according to your kind suggestions. Find the response below.

  1. “As a result, molecular oxygen was reduced to superoxide radicals as well as hydroxide radicals generated by the reaction of hole and molecules of water.” It must be the hydroxyl radicals not the hydroxide radicals on page 24 and also it must be corrected in Figure 13.

Response: Thank you so much for your positive comment. We have corrected this mistake in the revised version with new figure.

  1. In tables 2, 4, and 5, there is no removal efficiency. In fact, what is mentioned and measured in mg/g is called adsorption capacity.

Response: Thank you so much for your positive comment. We have corrected this mistake in the revised version.

Reviewer 4 Report

The authos solved the issues accordingly, and the manuscritp was greatly improved after revision. The manuscript can be accepted in current form. 

Author Response

Thank you so much for your positive comment and recommendation of the review article for publication in the present form.